# Translation attenuation by minocycline enhances longevity and proteostasis in old post-stress-responsive organisms

Gregory M Solis[1,2], Rozina Kardakaris[1], Elizabeth R Valentine[3,4], Liron Bar-Peled[1,5], Alice L Chen[1,5], Megan M Blewett[1,5], Mark A McCormick[6†], James R Williamson[3,4], Brian Kennedy[6], Benjamin F Cravatt[1,5], Michael Petrascheck[1,2]*

[1]Department of Molecular Medicine, The Scripps Research Institute, La Jolla, United States; [2]Department of Neuroscience, The Scripps Research Institute, La Jolla, United States; [3]Department of Integrative Structural and Computational Biology, The Skaggs Institute for Chemical Biology, The Scripps Research Institute, La Jolla, United States; [4]Department of Chemistry, The Skaggs Institute for Chemical Biology, The Scripps Research Institute, La Jolla, United States; [5]The Skaggs Institute for Chemical Biology, The Scripps Research Institute, La Jolla, United States; [6]The Buck Institute for Research on Aging, Novato, United States

*For correspondence:
pscheck@scripps.edu

Present address: [†]Department of Biochemistry and Molecular Biology, School of Medicine, University of New Mexico Health Sciences Center, New Mexico, United States

**Abstract** Aging impairs the activation of stress signaling pathways (SSPs), preventing the induction of longevity mechanisms late in life. Here, we show that the antibiotic minocycline increases lifespan and reduces protein aggregation even in old, SSP-deficient *Caenorhabditis elegans* by targeting cytoplasmic ribosomes, preferentially attenuating translation of highly translated mRNAs. In contrast to most other longevity paradigms, minocycline inhibits rather than activates all major SSPs and extends lifespan in mutants deficient in the activation of SSPs, lysosomal or autophagic pathways. We propose that minocycline lowers the concentration of newly synthesized aggregation-prone proteins, resulting in a relative increase in protein-folding capacity without the necessity to induce protein-folding pathways. Our study suggests that in old individuals with incapacitated SSPs or autophagic pathways, pharmacological attenuation of cytoplasmic translation is a promising strategy to reduce protein aggregation. Altogether, it provides a geroprotecive mechanism for the many beneficial effects of tetracyclines in models of neurodegenerative disease.

**Editorial note:** This article has been through an editorial process in which the authors decide how to respond to the issues raised during peer review. The Reviewing Editor's assessment is that all the issues have been addressed (see decision letter).

DOI: https://doi.org/10.7554/eLife.40314.001

## Introduction

Pharmacologic activation of longevity mechanisms to increase stress resistance and improve proteostasis capacity appears to be an attractive treatment strategy for degenerative diseases (*Balch et al., 2008*). While the exact mechanism(s) of neuronal death in Alzheimer's disease, Parkinson's disease and related maladies remain elusive, there is compelling genetic evidence for an age-associated collapse of proteostasis that contributes to protein aggregation and degenerative phenotypes (*Taylor and Dillin, 2011*). Most longevity mechanisms crucially depend on the ability to activate stress signaling pathways (SSPs) and proteostatic mechanisms (*Steinkraus et al., 2008*; *Tullet et al., 2008*; *Henis-Korenblit et al., 2010*; *Lapierre et al., 2013*). For example, a

comprehensive study in *Caenorhabditis elegans* by Shore et al. showed that the induction of SSPs is central and necessary for inducing longevity by 160 different RNAis (*Shore et al., 2012*). Others have shown that as animals age, their ability to respond to stress dramatically declines (*Labbadia and Morimoto, 2015*; *Dues et al., 2016*). As a consequence, longevity pathways whose mechanisms depend on the activation of SSPs can be predicted to become nonresponsive with aging.

Indeed, most longevity mechanisms, like the reduction of insulin/IGF signaling or a reduction in electron transport chain activity, do not extend lifespan when initiated past day 5 of *C. elegans* adulthood (*Dillin et al., 2002a*; *Dillin et al., 2002b*; *Rangaraju et al., 2015*). Proper timing is also required in Ames Dwarf mice, where growth hormone deficiency during the first 6 weeks of pre-pubertal development is critical for longevity (*Panici et al., 2010*).

Ideally one would like to be able to enhance proteostasis capacity and to extend life- and health-span by pharmacologically treating older organisms at the first appearance of neurodegenerative symptoms or disease biomarkers (e.g. protein aggregates). Fortunately, some longevity mechanisms such as dietary restriction or rapamycin treatment remain responsive later in life (*Weindruch and Walford, 1982*; *Harrison et al., 2009*). However, little is known about what distinguishes timing-specific longevity mechanisms from longevity mechanisms that remain responsive throughout life.

In this study, we identified minocycline, a drug known to have neuroprotective and anti-inflammatory properties in mammals (*Mitra et al., 1975*; *Choi et al., 2005*; *Festoff et al., 2006*; *Seabrook et al., 2006*; *Choi et al., 2007*; *Noble et al., 2009*; *Zheng et al., 2010*; *Cai et al., 2011*; *Ferretti et al., 2012*; *Obici et al., 2012*; *Cai et al., 2013*; *Garrido-Mesa et al., 2013*), to extend lifespan and to reduce protein aggregation even in old *C. elegans* that have lost their capacity to activate SSPs. Subsequent elucidation of the mechanism of action (MOA) shows minocycline to attenuate cytoplasmic translation independently of the integrated stress response (ISR), but mimicking its beneficial effects. We propose that the neuroprotective and aggregation-preventing effects of minocycline, observed in preclinical mouse models as well as in human clinical trials, are explained by its attenuation of cytoplasmic protein synthesis. Reducing the concentration of newly synthesized aggregation-prone proteins relieves demands on the proteostasis network even in older individuals in which SSPs and protein degradation pathways have already been compromised due to advanced age.

## Results

### Minocycline extends *C. elegans* lifespan in both young and old animals

Previous studies have shown that the capacity for *C. elegans* to respond to stress declines with age (*Bansal et al., 2015*; *Labbadia and Morimoto, 2015*). To expand on previous studies and to define the age at which major SSPs lose their capacity to respond to stress, we exposed transcriptional GFP-reporter strains that respond to oxidative stress (*gst-4p::GFP*) or unfolded proteins in the endoplasmic reticulum (*hsp-4p::GFP*, UPR$^{ER}$), in the mitochondria (*hsp-6p::GFP*, UPR$^{mt}$), or in the cytosol (*hsp-16.2p::GFP*) to their respective stressors -arsenite, tunicamycin, paraquat or heat- at increasing ages. On the first day as reproductive adults, herein defined as day 1, stressors increased GFP expression relative to untreated controls (*Figure 1A,B*). The exception was *hsp-6p::GFP* which was strongly activated at the L2 larval stage (*Figure 1—figure supplement 1A*) but responded poorly on day 1. By day 5 of adulthood, SSP activity was reduced and by day 8 was nearly undetectable (*Figure 1A,B*). Thus, 8 days after reaching adulthood *C. elegans* no longer induce SSPs in response to stress despite dying from its negative effects. We will refer to day 8 as the 'post-stress-responsive' age.

We next asked whether longevity and proteostasis mechanisms exist that remain drugable in post-stress-responsive adults. To this end, we treated 8-day-old *C. elegans* adults with 21 different molecules that we previously identified to extend lifespan (*Ye et al., 2014*). All 21 molecules extended lifespan when treatment was initiated on day 1 of adulthood. However, only minocycline extended lifespan when treatment was initiated on day 8 (*Figure 1C*; *Figure 1—figure supplement 1B*). For some of the inactive drugs, a xenobiotic-responsive *cyp-34A9p::GFP* strain was used to confirm drug uptake in old *C. elegans* (*Figure 1—figure supplement 1C*) (*Anbalagan et al., 2012*).

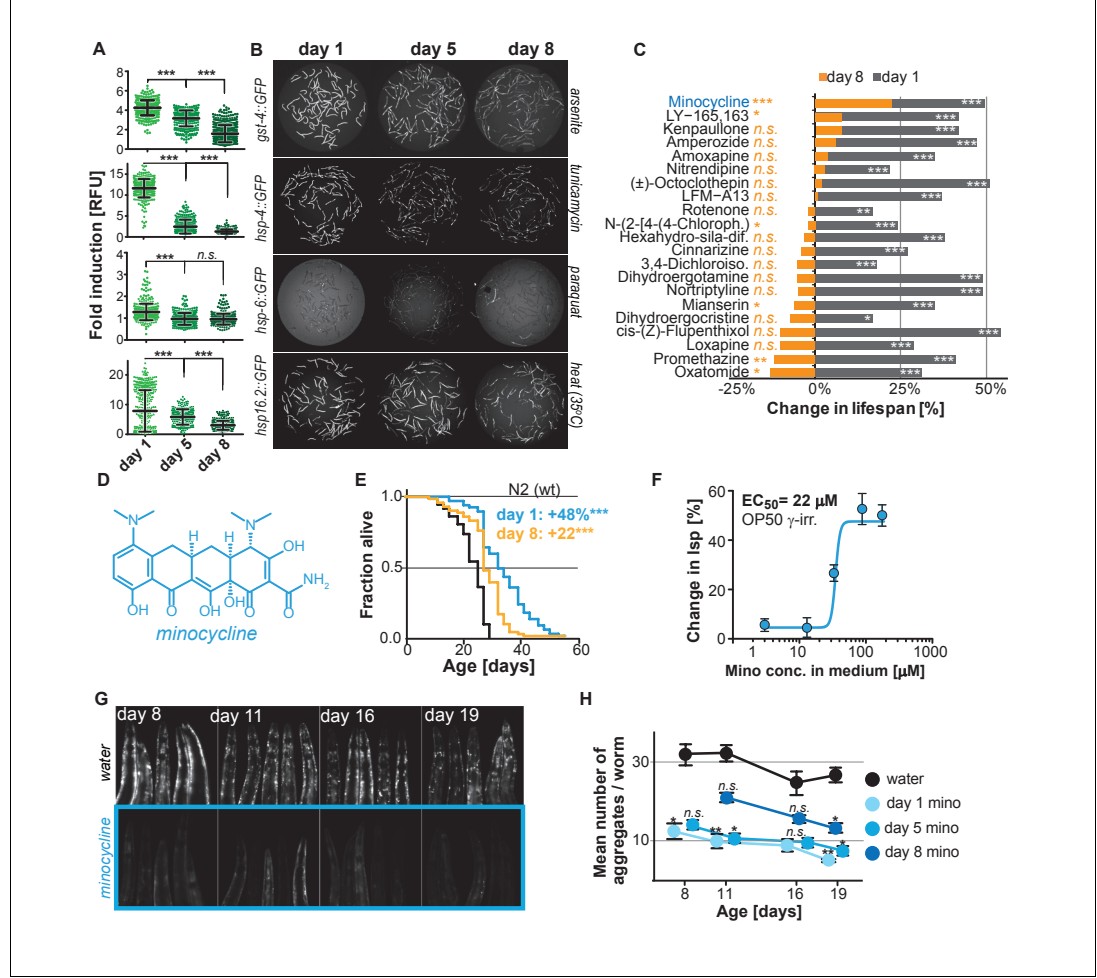

**Figure 1.** Stress signaling pathway activity declines with age. (**A**) Scatter plots show fold induction of GFP fluorescence induced by stressors compared to untreated animals in *gst-4p::GFP* (oxidative stress), *hsp-4p::GFP* (UPR^ER), *hsp-6p::GFP* (UPR^mt) and *hsp-16.2p::GFP* (heat stress) *C. elegans* at 1, 5 and 8 days of age. We define day 1 as the first day of adulthood. Error bars show mean ± S.D., each dot represents one animal with all n > 300 for each reporter strain and age. Significance determined by ANOVA followed by Dunett's test. Total of four independent experiments. (**B**) Representative fluorescent microscopy images of 100 randomly selected animals for each condition and strain. Stressors indicated to right of each image panel. Images for each strain were taken in parallel on the same day using identical settings. Total of four independent experiments. (**C**) Bar graphs show % change in lifespan for 21 small molecules when treatment was initiated on day 1 of adulthood (gray) or on day 8 of adulthood (orange). Note that some compounds that extend lifespan when treatment is initiated on day 1 become slightly toxic at later ages. One independent experiment with at least 53 animals per condition. Significance determined by the log-rank test. (**D**) Structure of minocycline. (**E**) Survival curves for untreated (black line) and minocycline-treated wild-type N2 animals when treatment was initiated on day 1 (blue line) or day 8 (orange line) of adulthood. OP50 were killed by γ-irradiation to separate antibiotic from lifespan effects. Total of five independent experiments. Significance determined by the log-rank test. (**F**) Percent change in lifespan as a function of minocycline concentration for N2. OP50 were γ-irradiated. Total of four independent experiments. (**G**) Representative fluorescence microscopy images showing heads of *C. elegans* at increasing ages expressing the α-synuclein::YFP fusion protein. Top: water-treated control, bottom: minocycline-treated (100 µM). Note the increase in punctuate staining with age indicative of protein aggregation. (**H**) Graph shows the average number of α-synuclein::YFP aggregates per worm as a function of age for water- or minocycline-treated animals. Age when minocycline treatment is initiated for each color is shown right of the graph. Error bars indicate S.E.M. For each data point, n > 15. Total of four independent experiments. Asterisks indicate significance *<0.05, **<0.01, ***<0.001, n.s. not significant. Source data for N2 lifespan experiments are available in the ***Figure 1—source data 1***.

DOI: https://doi.org/10.7554/eLife.40314.002

*Figure 1 continued on next page*

*Figure 1 continued*

The following source data and figure supplement are available for figure 1:

**Source data 1.** Summary of N2 lifespan data, related to *Figure 1*.

DOI: https://doi.org/10.7554/eLife.40314.004

**Figure supplement 1.** Small molecules screened for post-stress-responsive lifespan.

DOI: https://doi.org/10.7554/eLife.40314.003

Minocycline is a regulatory agency-approved tetracycline antibiotic used to treat acne vulgaris and has long been known to reduce tumor growth, inflammation and protein aggregation in mammals by an unknown MOA (*Figure 1D*) (*Garrido-Mesa et al., 2013*). Minocycline still extended lifespan when treatment was initiated on day 8, albeit by 22% instead of the 48% observed when treatment commenced on day 1 (*Figure 1E*). Subsequent control experiments showed that the lifespan extension by minocycline was independent of the method of sterilization, cultivation temperature (*Figure 1—figure supplement 1D,E*) and whether the animals were fed live or dead bacteria (OP50). Recording dose response curves we determined an $EC_{50}$ of 22 μM and an optimal concentration of 50 – 100 μM, as compared to therapeutic concentrations measured in patient serum of 5 – 10 μM (*Figure 1F*; *Figure 1—figure supplement 1E*) (*Agwuh and MacGowan, 2006*; *Garrido-Mesa et al., 2013*). To investigate the MOA of minocycline independently of its antibiotic activity, all subsequent experiments were conducted using only dead, γ-irradiated bacteria. Taken together, minocycline extends lifespan independently of its antibiotic activity, even when treatment is initiated in old, post-stress-responsive *C. elegans*.

## Minocycline attenuates protein aggregation in both young and old *C. elegans*

We next tested minocycline for its ability to modulate protein aggregation, a common molecular characteristic of many neurodegenerative diseases. We treated animals that express human α-synuclein fused to YFP in the muscle (*van Ham et al., 2008*) with water or minocycline on day 1 of adulthood and imaged them on days 8, 11, 16 and 19. Minocycline suppressed the age-dependent increase in α-synuclein aggregation. It also reduced α-synuclein aggregation even when added to post-stress-responsive adults at a late age (day 8), when unfolded protein responses were no longer induced (*Figure 1G,H*; *Figure 1—figure supplement 1F*). To confirm this effect extended beyond α-synuclein aggregation, we tested another model of protein aggregation and determined minocycline also reduced the paralysis caused by temperature-induced $A\beta_{1-42}$ misfolding that leads to aggregation in *C. elegans*' body-wall muscle (*Figure 1—figure supplement 1G*) (*Jiang et al., 2012*; *McColl et al., 2012*). Thus, minocycline treatment reduced both age-dependent and temperature-induced protein aggregation in *C. elegans*.

## Minocycline extends *C. elegans* lifespan in mutants defective for SSP or autophagy activation

As minocycline extends lifespan of post-stress-responsive *C. elegans*, its activity should be independent of SSP activation. This prediction was tested in two ways. First, we repeated the GFP reporter-based stress response activation experiments, pretreating day-5-old adults for 2 hr with minocycline before subjecting them to stress (*Figure 2A*). As expected, minocycline alone did not induce any SSP reporter (*Figure 2B,C*). Unexpectedly, pretreatment with minocycline suppressed stressor-induced activation of all SSP reporters compared to stressor alone. This was particularly surprising as previous studies have shown that treating larvae with minocycline activates the *hsp-6p::GFP* reporter (*Figure 2—figure supplement 1A,B*) (*Houtkooper et al., 2013*). In contrast, in adults, minocycline inhibited the activation of all SSPs. Minocycline-induced inhibition of SSPs did not result in an increased susceptibility to stress. Minocycline-treated adults survived heat stress (35°C) much better than untreated adults, with 60% of the minocycline-treated animals still alive when nearly all control animals had died (*Figure 2—figure supplement 1C*). Furthermore, minocycline-treated adults were also much more resistant to paraquat-induced oxidative stress (*Figure 2—figure supplement 1D*). This is consistent with the observation that minocycline also protects neuronal-like rat pheochromocytoma (PC12) cells from paraquat-induced cell death (*Huang et al., 2012*). Thus, despite inhibiting

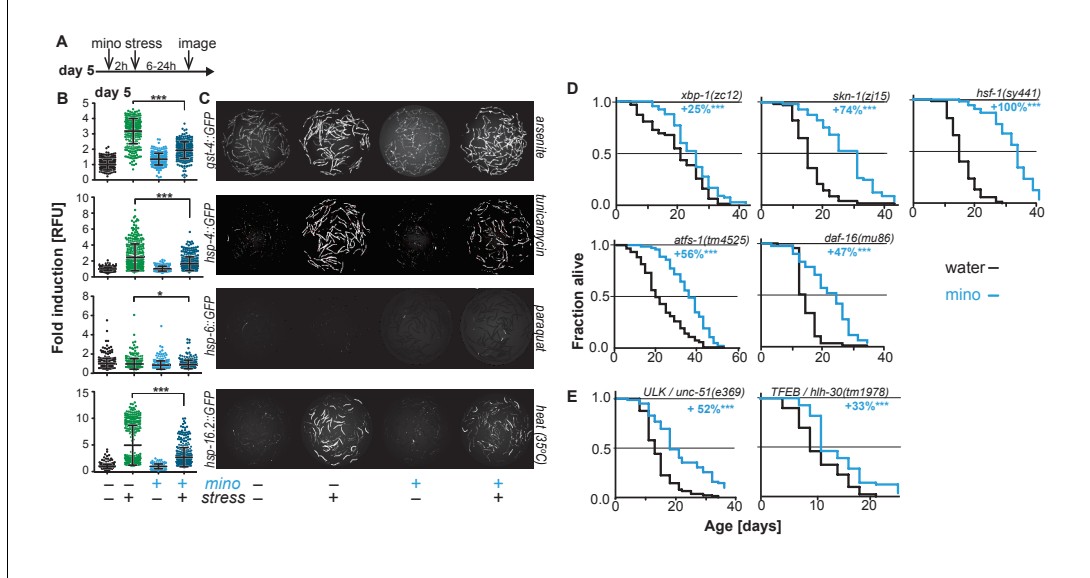

**Figure 2.** Minocycline suppresses stress signaling pathway activity. (A) Experimental timeline to monitor stress response activation of 5-day-old adult *C. elegans* GFP-reporter strains induced by stressor and/or minocycline. Minocycline (100 µM) was added 2 hr prior to each stressor. (B) Scatter plots show fold induction of GFP fluorescence induced by stressors and/or minocycline treatment compared to untreated animals in 5-day-old adults. Pretreatment with minocycline suppresses stress response activation. Reporters: *gst-4p::GFP* (oxidative stress), *hsp-4p::GFP* (ER UPR), *hsp-6p::GFP* (UPRmt) and *hsp-16.2p::GFP* (heat stress). Error bars show mean ± S.D., each dot represents one animal with all n > 300. Significance determined by the Mann-Whitney t-test. At least three independent experiments. (C) Representative fluorescence microscopy images of 100 randomly selected animals for each condition and strain. Stressors indicated to the right of each panel. Images for each strain were taken in parallel on the same day using identical settings. (D) Survival curves. Minocycline significantly extends lifespan in strains carrying mutations in regulators of stress and proteostasis responses. Statistical significance determined by the log-rank test. Number of animals n ranging from 35 to 218. Total of at least three independent experiments per strain. (E) Survival curves. Minocycline significantly extends lifespan in two strains carrying mutations in regulators of lysosomal and autophagic pathways. Statistical significance determined by the log-rank test. Number of animals n ranging from 56 to 138. Total of at least three independent experiments per strain. Asterisks indicate significance *<0.05, **<0.01, ***<0.001, n.s. not significant. Source data for all lifespan experiments in (D) and (E) are available in the *Figure 2—source data 1*.

DOI: https://doi.org/10.7554/eLife.40314.005

The following source data and figure supplement are available for figure 2:

**Source data 1.** Summary of lifespan data for strains carrying mutations in regulators of stress, proteostasis, autophagy and lysosomal responses, related to *Figure 2*.

DOI: https://doi.org/10.7554/eLife.40314.007

**Figure supplement 1.** Despite inhibiting the UPRmt and the heat shock response in adults, minocycline activates the UPRmt in larvae and protects from stress.

DOI: https://doi.org/10.7554/eLife.40314.006

SSP activation, minocycline protects from stress, suggesting a protective mechanism that bypasses SSP activation.

Second, we measured the lifespan of water- and minocycline-treated strains carrying mutations in genes encoding transcription factors that regulate SSP activity, including the FOXO transcription factor *daf-16*, the UPRER transcriptional activator *xbp-1*, the UPRmt transcriptional activator *atfs-1*, the oxidative stress response factor *skn-1*, and the universally conserved heat shock transcription factor *hsf-1*. Minocycline extended lifespan in all *C. elegans* mutants (*Figure 2D*). Remarkably, minocycline extended lifespan in *hsf-1(sy441)* mutants by up to +159%, a relative extension clearly greater than what was observed in wild-type N2 animals and almost reaching absolute lifespans of minocycline-treated N2 animals (*Figure 1E*; *Figure 2—source data 1*). This enhanced extension suggests that minocycline not only extends lifespan as in N2 but also rescues some of the defects associated with mutated *hsf-1(sy441)*.

One way how minocycline could rescue *hsf-1*-associated defects would be to clear misfolded proteins by inducing autophagy and lysosomal pathways (*Kumsta et al., 2017*). To determine if autophagy is required for the lifespan extension by minocycline, we measured lifespan in minocycline-

treated strains harboring mutations in the Atg1/ULK1 ortholog *unc-51* and the TFEB ortholog *hlh-30*, key factors critical for autophagy and lysosomal activity. In both strains, minocycline significantly increased lifespan (*Figure 2E*). Despite the severe behavioral and morphological phenotypes observed in the *unc-51(e369)* mutants, the mutation did not reduce the ability of minocycline to extend lifespan. Minocycline also extended the lifespan of *hlh-30(tm1978)* mutants, but less than the 45% observed in N2. These genetic results, and the previous finding that activation of autophagy in already old *C. elegans* is likely to be detrimental (*Wilhelm et al., 2017*), are consistent with a mechanism of minocycline that acts independently of the activation of autophagy. However, as *unc-51* is essential, the residual activity remaining in the *unc-51(e369)* mutants does not allow us to draw a definitive conclusion.

## Minocycline reduces reactive cysteine labeling at the ribosome

In contrast to minocycline, lifespan extension induced in *C. elegans* by the conserved longevity paradigms including dietary restriction, reduced mitochondrial activity, germline ablation, sensory perception, reduced insulin signaling or the inhibition of mTOR all depend on at least one of the above tested factors (*Steinkraus et al., 2008*; *Robida-Stubbs et al., 2012*; *Houtkooper et al., 2013*; *Lapierre et al., 2013*; *Weir et al., 2017*). To gain insight into the MOA of minocycline, we conducted activity-based protein profiling using iodoacetamide (IA) as a probe, followed by isotopic tandem orthogonal proteolysis (isoTOP-ABPP). IA binds specifically to reactive cysteine residues, including catalytic sites within enzymes, post-translational modification sites, cysteine oxidation sites, and other types of regulatory or functional domains across the proteome (*Roberts et al., 2017*). We hypothesized that minocycline treatment would directly, by binding near a reactive cysteine, or indirectly, by modulating the activity of a pathway, alter IA labeling of reversible, post-translational cysteine modifications.

To determine minocycline-induced changes in cysteine labeling patterns, 5-day-old adult *C. elegans* were treated with water or minocycline for 3 days and their proteome was subsequently labeled with the IA probe and analyzed by mass spectrometry (*Figure 3—figure supplement 1A*). Analysis of the isoTOP-ABPP data by the Cytoscape plugin ClueGO (*Bindea et al., 2009*), which allows visualization of biological terms for large clusters of genes in a functionally grouped network, showed that minocycline treatment reduced IA labeling of proteins involved in cytoplasmic translation, specifically proteins involved in ribosome assembly and peptide metabolic processes (*Figure 3A*). These results suggested that minocycline directly or indirectly targets cytoplasmic translation, an MOA consistent with lifespan extension and that explained how minocycline prevented SSP reporter induction (*Figure 2B,C*) (*Vellai et al., 2003*; *Kapahi et al., 2004*; *Kaeberlein et al., 2005*; *Hansen et al., 2007*; *Pan et al., 2007*; *Syntichaki et al., 2007*; *McQuary et al., 2016*).

## Minocycline attenuates cytoplasmic mRNA translation in *C. elegans*

We next set out to determine the effect of minocycline treatment on cytoplasmic mRNA translation. Consistent with reducing translation, minocycline treatment reduced *C. elegans* growth and reproduction (*Figure 3B,C*; *Figure 3—figure supplement 1B,C*) (*Hansen et al., 2007*; *Pan et al., 2007*). To quantify mRNA translation in the presence or absence of minocycline, we measured incorporation of $^{14}N$ and $^{15}N$-labeled amino acids into the proteome by quantitative mass spectrometry. In a first series of experiments, synchronized populations of animals at the L4 larval stage were treated for 24 hr with either water or minocycline and fed $^{14}N$-labeled bacteria. To quantify differences in protein levels, a $^{15}N$ standard was added to determine a $^{14}N$ to ($^{14}N + ^{15}N$) intensity ratio. The majority of proteins (85%) in the minocycline-treated samples showed a significant reduction in intensity compared to controls, revealing that minocycline treated animals contained ~24% less protein (*Figure 3D*,L4→ day 1). These effects on the proteome were not likely the result of increased protein degradation, as minocycline reduced rather than increased proteosomal activity (*Figure 3—figure supplement 1D*). At least in adult *C. elegans* these data were inconsistent with a model in which minocycline selectively reduced translation, as has been shown for *ifg-1* (*Rogers et al., 2011*). The few factors that showed higher expression levels in minocycline-treated animals had no known link to stress resistance or were shown to increase lifespan when knocked down by RNAi.

In a second series of experiments, we made use of a previous observation showing that mRNA translation rates in *C. elegans* decline with age (*Gomez-Amaro et al., 2015*). Thus, if minocycline

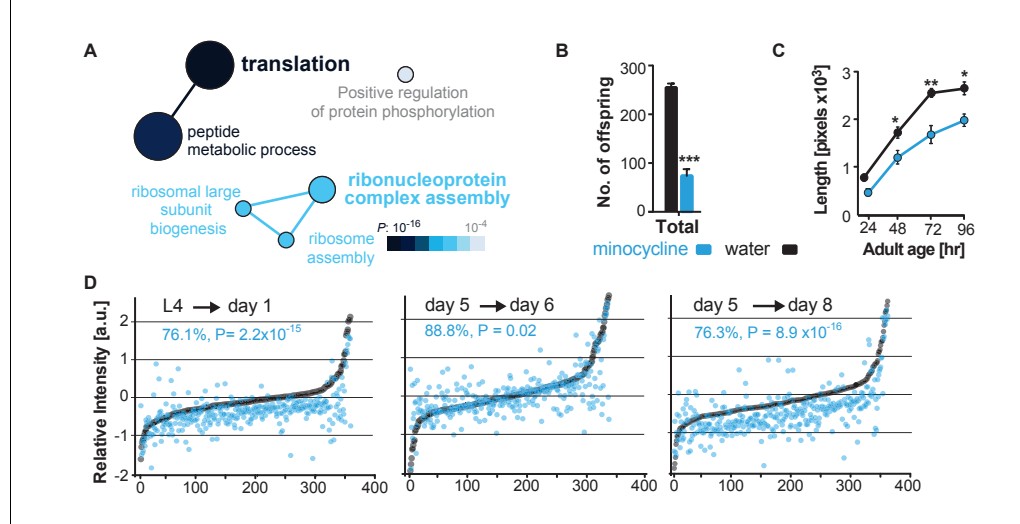

**Figure 3.** Minocycline attenuates translation in *C. elegans*. (**A**) Network analysis by ClueGO of isoTOP-ABPP results corresponding to protein-probe labeling changes that decrease with minocycline treatment. Size of the circle is proportional to the number of proteins identified and the color represents significance. Total of three independent experiments. (**B**) Graph shows total number of offspring for water- or minocycline-treated *C. elegans* over 5 days after minocycline addition. Significance determined by the Mann-Whitney t test n = 15. Total of five independent experiments. (**C**) Graph shows *C. elegans* length in pixels as a function of age in hours after minocycline addition. Significance determined by the Mann-Whitney Student's t test n > 50. Total of five independent experiments. (**D**) Graphs show relative intensity ratios of $^{14}N$ incorporation to total $^{14}N +^{15}N$ -labeled proteins in water (black circles) or minocycline-treated (blue circles) animals as a function of abundance in water-treated animals at different ages. Length and age of labeling indicated above each graph. Total of two independent experiments. Significance determined by a paired t test. Asterisks indicate significance *<0.05, **<0.01, ***<0.001, n.s. not significant. Source data for isoTOP-ABPP and quantitative mass spectrometry experiments are available in *Figure 3—source datas 1* and *2*, respectively.

DOI: https://doi.org/10.7554/eLife.40314.008

The following source data and figure supplement are available for figure 3:

**Source data 1.** Summary of isoTOP-ABPP analysis.
DOI: https://doi.org/10.7554/eLife.40314.010
**Source data 2.** Summary of $^{15}N$-incorporation analysis.
DOI: https://doi.org/10.7554/eLife.40314.011
**Figure supplement 1.** Minocycline-treated animals display phenotypes characteristic of translation inhibition, Related to *Figure 3*.
DOI: https://doi.org/10.7554/eLife.40314.009

acted by attenuating translation, its effect on $^{14}N$ incorporation into the proteome should depend on the translation rate and should be smaller in 5-day-old adult animals than in young, L4 larvae. To test this prediction, we initiated minocycline treatment on day 5 of adulthood and harvested protein 24 hr and 72 hr later. As predicted, the effect of minocycline on mRNA translation depended on the age of the animals. While a 24 hr minocycline treatment initiated in L4 larvae reduced protein synthesis by ~24% (*Figure 3D,L4*→day 1), the same length of treatment initiated in day-5-old adults showed only an ~11% reduction on mRNA translation (*Figure 3D*, day 5→day 6). However, a 72-hr minocycline treatment initiated on day 5 allowed enough mRNA translation to occur to observe a significant suppression of ~24% (*Figure 3D*, day 5→day 8). In addition, we observed the suppressive effect of minocycline to correlate with protein abundance, suppressing the production of highly expressed proteins to a greater extent than lowly expressed proteins (*Figure 3—figure supplement 1E*). Taken together, these results show that minocycline-treated animals contain less protein, consistent with an attenuation of mRNA translation.

## Minocycline attenuates mRNA translation in human cells

To determine if minocycline treatment also can reduce mRNA translation in human cells, we measured basal mRNA translation by monitoring incorporation of [$^{35}$S]-methionine/cysteine in untreated and minocycline-treated HeLa cells. A 6 hr pretreatment with minocycline reduced mRNA translation by ~30% at 100 μM as monitored by [$^{35}$S]-methionine/cysteine incorporation (*Figure 4A*; *Figure 4—figure supplement 1A*). This experiment was repeated in murine NIH 3T3 cells with similar results (*Figure 4—figure supplement 1B*). The magnitude of the effect was similar to what we observed in *C. elegans* (*Figure 3D*), but less than the effect of cycloheximide (*Figure 4A*). As the bioavailability of minocycline is close to 100%, *in vivo* and *in vitro* concentrations are expected to be similar (*Garrido-Mesa et al., 2013*).

To gain further insight on how minocycline reduces mRNA translation, we recorded polysome profiles from untreated and minocycline-treated HeLa cells. Minocycline treatment caused a pronounced increase in the 80S monosome peak (M) indicative of stalled translation initiation (*Figure 4B*). It further showed a disproportionally strong reduction in the heavy polysome fraction (P), suggesting that minocycline reduced ribosomal load, the number of ribosomes per mRNA, of highly translated mRNAs. Thus, the observed reduction of ribosomal load by minocycline results in a substantial attenuation of highly translated mRNAs. The preferential effect on the heavy polysome fraction (P) provided a potential explanation for the greater inhibitory effect of minocycline on highly expressed proteins that we observed in the *C. elegans* metabolic labeling experiment (*Figure 3—figure supplement 1E*).

Upon heat shock, polysome-mediated translation allows cells to quickly synthesize large quantities of chaperones like HSP60 and HSP70. If minocycline attenuates polysome-mediated translation it should substantially suppress the heat shock-induced expression of HSP60 and HSP70 at the protein but not at the mRNA level. To test this, we pretreated HeLa cells with water or minocycline and subjected them to heat shock to measure HSP60 and HSP70 protein expression levels by western blot. As expected, HeLa cells strongly induced expression of HSP60 and HSP70 in response to a heat shock while pretreatment with minocycline dramatically reduced it (*Figure 4C*). However, minocycline only reduced HSP60 and HSP70 expression at the protein level and had no effect on the induction of their mRNAs (*Figure 4D*). Similarly, we revisited *C. elegans* and confirmed that minocycline suppressed the activation of *hsp-16.2p::GFP* at the protein but not at the mRNA level (*Figure 2B*; *Figure 4—figure supplement 1C*). Thus, minocycline-treated cells activate stress responses at the level of mRNA but are unable to rapidly translate them into proteins as minocycline strongly attenuates polysome-mediated translation.

To extend these results to other stress responses such as the UPR$^{ER}$, we repeated these experiments using HEK293$^{DAX}$ cells. HEK293$^{DAX}$ cells allow ligand-induced activation of the UPR$^{ER}$ in the absence of stress, as they express the UPR$^{ER}$ transcriptional activator ATF6 fused to mutant destabilized dihydrofolate reductase (DHFR). Because the DHFR domain is largely unfolded, the ATF6-DHFR fusion protein is continually degraded by the proteasome. The addition of trimethoprim (TMP) stabilizes the ATF6-DHFR fusion, leading to a dose-dependent expression of the chaperone BiP (*Shoulders et al., 2013*). As expected, TMP treatment dramatically induced BiP expression that was completely suppressed by minocycline co-treatment (*Figure 4E*; *Figure 4—figure supplement 1D*). These results show that minocycline strongly suppresses activation of multiple stress responses at the level of translation in both *C. elegans* and human cells irrespective of whether expression was induced by stress or an artificial ligand like TMP.

As we did before in *C. elegans*, we asked if the suppression of stress responses by minocycline impaired the ability of a cell to deal with heat stress and protein aggregation (*Figure 1—figure supplement 1G*; *Figure 2—figure supplement 1C*). Using the aggregation-specific dye ProteoStat, we measured protein aggregation in HeLa cells before and after a 1 hr heat shock. The protein aggregate-induced fluorescence only increased in control cells upon heat shock (~12%), but not in minocycline-pretreated cells (*Figure 4F*). Thus, as already observed in *C. elegans*, minocycline treatment protects from heat-shock-induced protein aggregation despite suppressing activation of the heat-shock response (*Figure 1—figure supplement 1G*; *Figure 2—figure supplement 1C*).

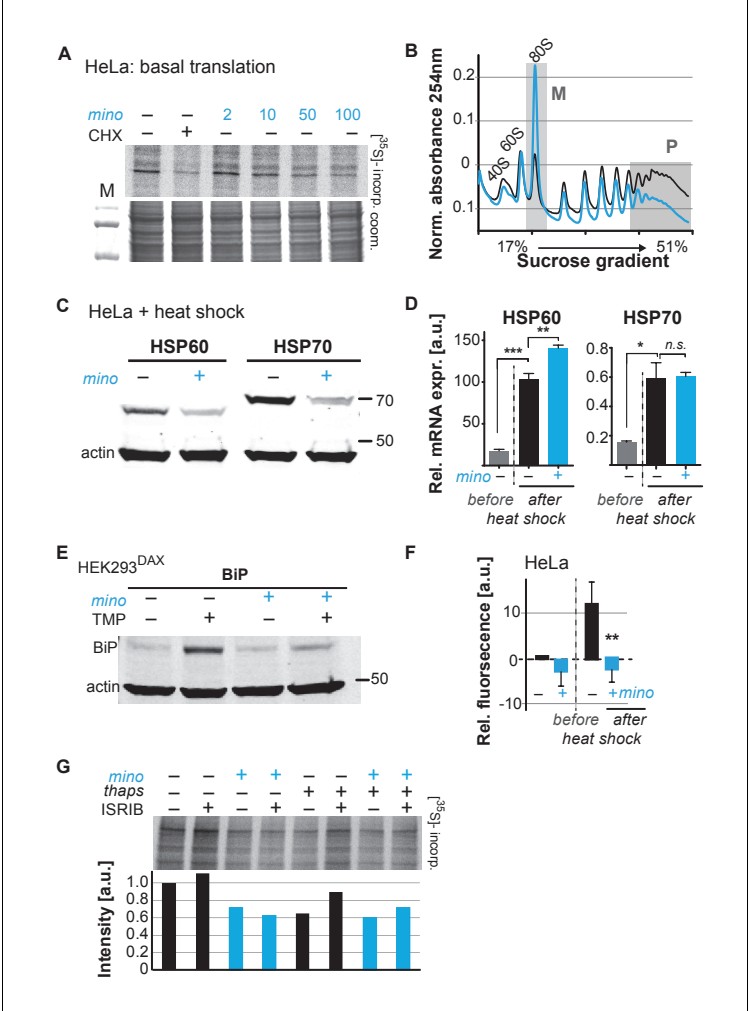

**Figure 4.** Minocycline attenuates translation and reduces ribosomal load of highly translated mRNAs. (**A**) Autoradiograph (top) monitoring $^{35}$S incorporation over 1 hr of HeLa cells treated for 6 hr with cycloheximide or increasing concentrations of minocycline. Coomassie gel (bottom) is shown as a loading control. Total of four independent experiments. (**B**) Polysome profile analysis of 12 hr water- (black) or minocycline-treated (blue) HeLa cells. High molecular weight polysomes (**P**) and 80S monosomes (**M**) regions are shaded in gray. An increase in monosomes and a decrease in high-molecular-weight polysomes are indicative of translation attenuation. Total of three independent experiments. (**C**) Immunoblots probing for HSP60 and HSP70 expression of water- or minocycline-treated (100 μM) HeLa cells after a 1 hr, 43°C heat shock. Total of three independent experiments. (**D**) qRT-PCR analysis of HSP60 and HSP70 mRNA expression in HeLa cells treated with or without minocycline (100 μM) before and after a 1 hr, 43°C heat shock. Minocycline treatment was initiated 12 hr prior to the heat shock. Data are represented as mean ± S.E.M. Significance determined by the Mann-Whitney t-test. Total of three independent experiments. (**E**) Immunoblots probing for BiP expression of water- or minocycline-treated (100 μM) HEK293$^{DAX}$ cells engineered to allow UPR$^{ER}$ activation by trimethoprim treatment (TMP). Water or 100 μM minocycline was added alone or in combination with 20 uM trimethoprim for 12 hr prior to cell lysis. Samples were normalized by RNA concentration. Total of three independent experiments. (**F**) Bar graph shows change in protein aggregate formation, as measured by the ProteoStat assay, before and after a 1 hr, 43°C heat shock in control (black) and minocycline-treated (blue) HeLa cells. Data are represented as mean ± S.E.M. Significance determined by the Mann-Whitney t-test. Total of four independent experiments. (**G**) Autoradiograph (top) monitoring $^{35}$S incorporation of HeLa cells treated with the ISR inhibitor ISRIB (200 nM), minocycline (100 μM), thapsigargin (thaps, 200 nM) and a combination of the three. Coomassie gel (bottom) is shown as a loading control. Total of two independent experiments. Asterisks indicate significance *<0.05, **<0.01, ***<0.001, n.s. not significant. Figure 5. Lifespan extension by minocycline depends on cytoplasmic translation.
DOI: https://doi.org/10.7554/eLife.40314.012

The following figure supplement is available for figure 4:

*Figure 4 continued on next page*

*Figure 4 continued*

**Figure supplement 1.** Minocycline suppresses [35]S incorporation in multiple paradigms.
DOI: https://doi.org/10.7554/eLife.40314.013

## Minocycline attenuates translation independent of the ISR

SSP activation leads to the phosphorylation of the eukaryotic translation initiation factor eIF2α, globally inhibiting translation initiation of all but a few mRNAs, generally referred to as the integrated stress response (ISR) (*Pakos-Zebrucka et al., 2016*). Minocycline could either attenuate translation through activation of the ISR or by directly targeting the cytoplasmic ribosome to interfere with translation. To distinguish between those possibilities, we made use of the ISR inhibitor ISRIB that prevents eIF2α phosphorylation-mediated translational inhibition (*Sidrauski et al., 2015*). If minocycline activates the ISR to attenuate translation, co-treatment with ISRIB should restore normal translation even in the presence of minocycline. HeLa cells were treated with ISRIB, minocycline or both agents. Monitoring translation by [35S]-methionine/cysteine incorporation showed that co-treatment with ISRIB did not impair the ability of minocycline to attenuate translation (*Figure 4G*; *Figure 4—figure supplement 1E*). In contrast, translation attenuation by thapsigargin, a known inducer of the ISR, was restored by the co-treatment with ISRIB (*Sidrauski et al., 2015*). Thus, minocycline attenuates translation by an ISR-independent mechanism. Comparing the bacterial 16S rRNA to the cytoplasmic 18S rRNA of *C. elegans* and *H. sapiens* revealed a conserved tetracycline binding site in helix 34 (h34) (*Figure 4—figure supplement 1F*) (*Brodersen et al., 2000*). A recent paper by the Meyers group identified doxycycline to directly bind to key 18S rRNA substructures of the cytoplasmic ribosome and showed different tetracyclines to discriminate between different 18S rRNA binding sites (*Caballero et al., 2011*; *Mortison et al., 2018*). Thus, we concluded that minocycline directly targets the 18S rRNA, leading to the observed stalled ribosomes (80S peak) and attenuation of translation (*Figure 4A,B*).

## Minocycline extends lifespan by attenuating translation

Translation of mRNA is essential and 'null' mutants lacking 18S rRNA are not viable, preventing us from directly testing the necessity of mRNA translation for the minocycline-induced longevity. In contrast to our studies in *C. elegans* and human cells, studies investigating the effects of tetracyclic antibiotics in *S. cerevisiae* have shown them to only attenuate mitochondrial but not cytoplasmic translation. Consistent with a model in which minocycline needs to attenuate cytoplasmic translation to extend lifespan, minocycline did not extend replicative lifespan in *S. cerevisiae,* and at higher concentrations reduced it (*Figure 5A*) (*Clark-Walker and Linnane, 1966*; *Caballero et al., 2011*; *McCormick et al., 2015*).

To more directly link translation inhibition by minocycline to its effect on longevity, we reasoned that if minocycline extends lifespan by reducing translation directly at the ribosome, its dose-response curve should be shifted to the left in mutants with already reduced translation and shifted to the right in mutants with increased translation (*Figure 5B*).

For a mutant with reduced translation, we used the *rsks-1(ok1255)* strain (*Hansen et al., 2007*; *Pan et al., 2007*). The *rsks-1(ok1255)* strain carries a deletion in the homolog of the S6 kinase, a regulator of translation initiation. Genetic data have shown that *rsks-1* extends lifespan by a mechanism dependent on *hlh-30*, and thus different from the mechanisms underlying minocycline (*Lapierre et al., 2013*). By measuring protein concentrations by Bradford assay in young adults, prior to the generation of eggs, we further confirmed the independence of the two mechanisms. In *rsks-1* mutants, protein concentration was reduced by 18% (±8%) compared to N2. Treatment of *rsks-1 (ok1255)* mutants with minocycline had an additive effect and further reduced protein concentration by an additional 26%, resulting in a total reduction of 44% (±8%, p=0.0018, Dunett) compared to N2. Thus, minocycline and the *rsks-1* mutation additively reduced protein concentrations. For a mutant with increased translation, we used the *ncl-1(e1865)* strain (*Frank and Roth, 1998*). Strains carrying mutations in *ncl-1* express twice as much 18S rRNA and produce 22% more protein than wild-type animals. As our and the Mortison data (*Mortison et al., 2018*) suggest that minocycline directly binds to the *C. elegans* 18S rRNA, a two-fold increase in the 18S rRNA level and the

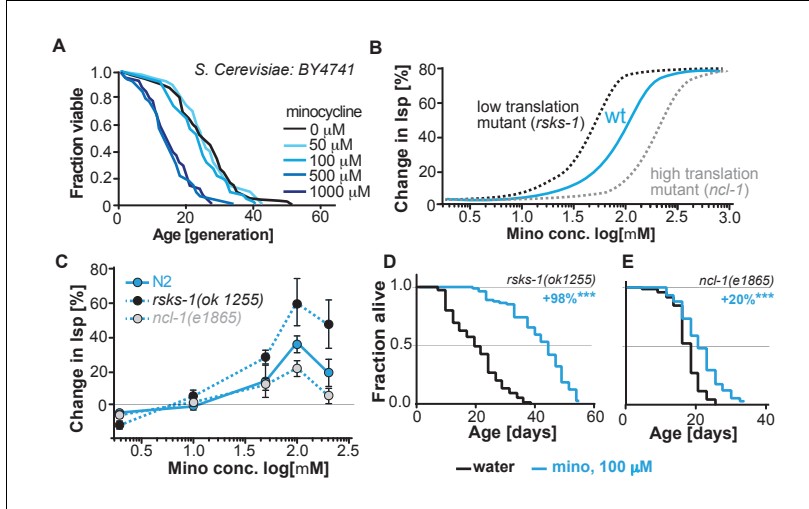

**Figure 5.** (A) Replicative lifespan of *S. cerevisae* treated with increasing concentrations of minocycline. Data were analyzed using the Wilcoxon rank-sum test. (B) Depicts expected dose response curve shifts for mutants with lower (black curve) or higher (gray curve) translation rates treated with minocycline compared to wild-type (blue curve). As translation is essential, translation mutants must retain some translation activity, albeit less than in wild type. Thus, in mutants with reduced translation like *rsks-1*, less minocycline should be necessary to optimally lower translation and to increase lifespan resulting in a left-shifted dose-response curve. If minocycline targets the 18S rRNA, an excess of rRNA and an increase in translation as in *ncl-1* mutants should result in a right-shifted dose-response curve. (C) Dose response curves show the % change in lifespan as a function of increasing minocycline concentrations for N2, *rsks-1(ok1255)* and *ncl-1(e1865)* mutants. Total of four independent experiments performed. Data are represented as mean ± S.E.M. (D) Survival curves for water- or minocycline- (100 µM) treated *rsks-1 (ok1255) C. elegans* mutants. At least four independent experiments performed. (E) Survival curves for water- or minocycline- (100 µM) treated *ncl-1(e1865) C. elegans* mutants. Statistical significance determined by the log-rank test. Number of animals n ranging from 42 to 87, total of four independent experiments. Asterisks indicate significance *<0.05, **<0.01, ***<0.001, n.s. not significant. Source data for N2, *rsks-1* and *ncl-1* lifespan experiments are available in ***Figure 5—source data 1***.

DOI: https://doi.org/10.7554/eLife.40314.014

The following source data and figure supplement are available for figure 5:

**Source data 1.** Summary of *rsks-1* and *ncl-1* lifespan data, related to ***Figure 5*** and ***Figure 5—figure supplement 1***.
DOI: https://doi.org/10.7554/eLife.40314.016
**Figure supplement 1.** Evidence for a toxic off-target at higher minocycline concentrations.
DOI: https://doi.org/10.7554/eLife.40314.015

associated increase in translation should shift the dose-response curve to the right and diminish minocycline's effect on lifespan (***Figure 5B***).

As depicted in ***Figure 5C***, lifespan extension by minocycline was reduced in *ncl-1(e1865)* mutants with a right-shifted dose-response curve (***Figure 5C,E***). In contrast, in long-lived *rsks-1(ok1255)* mutants, we observed a left-shifted dose-response curve (***Figure 5C,D***). Thus, less minocycline was required in *rsks-1(ok1255)* mutants to achieve the same level of lifespan extension compared to wild-type animals, as the translation levels are already reduced by the mutation. However, two aspects of the dose-response curves were unexpected. First, minocycline increased lifespan by an average of more than 60% and up to 98% in the already long-lived *rsks-1* mutant. Second, the dose-response curves for all three strains started to decline between 100 and 200 µM (***Figure 5—figure supplement 1***). In theory, this decline could be caused by too much translation inhibition, which at some point becomes detrimental. However, since all strains, irrespective of whether they translate at wild-type (N2) rates, lower rates (*rsks-1*) or higher rates (*ncl-1*) show this drop at the same concentration, this effect is unlikely the result of too much translation inhibition, but more likely the result of a toxic off-target (***Figure 5—figure supplement 1***).

## Discussion

Ameliorating pathological protein aggregation requires pharmacological mechanisms that improve proteostasis, even under circumstances when SSPs are compromised, as in aging. To identify such mechanisms, we searched for a geroprotective compound capable of extending lifespan and improving proteostasis in post-stress-responsive *C. elegans*. These efforts identified minocycline, a regulatory agency-approved antibiotic also known to reduce inflammation and protein aggregation in mice and humans by a hitherto unknown MOA (*Supplementary file 1*; *Supplementary file 2*). Using an unbiased chemo-proteomic approach, we identified the MOA by which minocycline extends lifespan and enhances proteostasis. Our ABPP data show that minocycline targets the ribosome (*Figure 3A*) to attenuate translation (*Figure 3D*; *Figure 4A*), preferentially reducing polysome formation (*Figure 4B*), mimicking the protective effects of the ISR without working through this pathway (*Figure 4G*; *Figure 6*). Notably, this MOA also provides a unifying explanation for the many other seemingly unrelated effects of minocycline observed in preclinical and clinical studies, including its ability to reduce tumor growth, inflammation, and improve symptoms of fragile X. In fact,

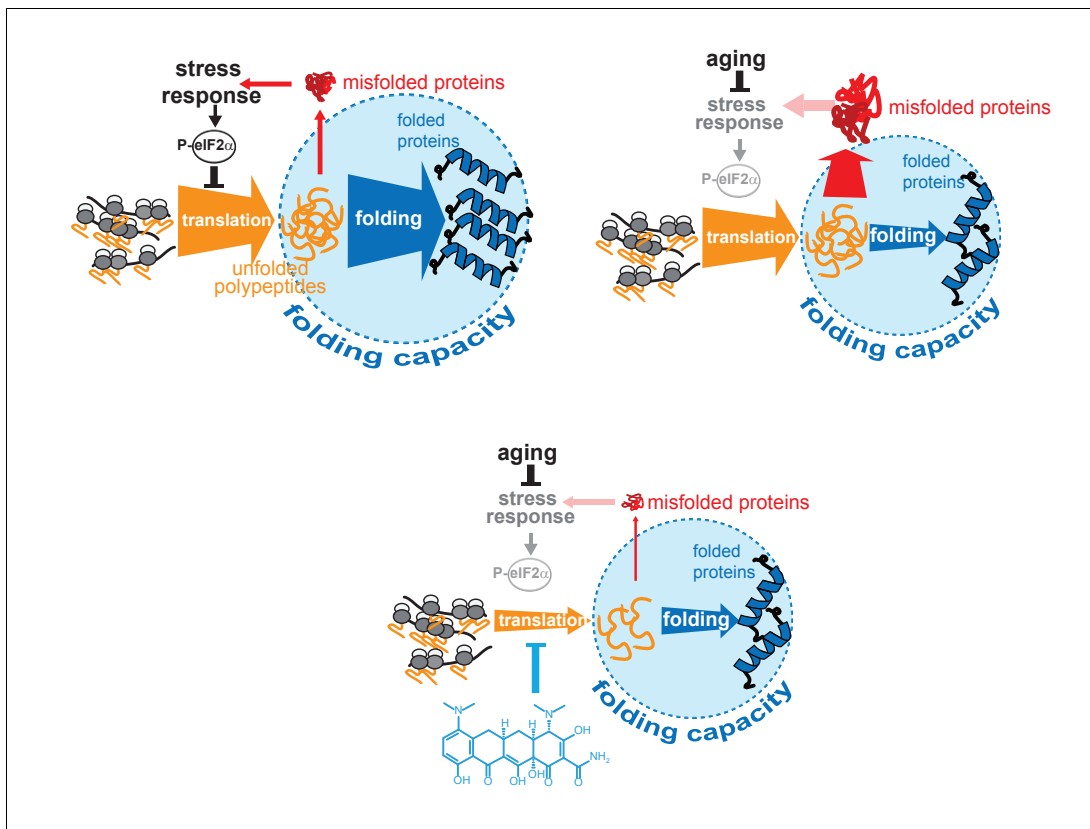

**Figure 6.** Model on how minocycline rebalances proteostatic load with the decreased folding capacity in older organisms. Top left: In young animals stress responses and the ISR act as feedback mechanisms to control proteostatic load driven by mRNA translation. Translation is prevented from overburdening the folding machinery as accumulation of unfolded proteins leads to the induction of stress responses, triggering attenuation of translation through eIF2α phosphorylation, acting as a feedback control to reduce protesostatic load. Top right: Age-associated decline of stress response signaling compromises the translational feedback attenuation, allowing a relative excess in mRNA translation compared to the declining folding capacity in older adults. Thus, in old organisms cellular signals (i.e. inflammatory signals) that induce widespread gene expression changes and protein synthesis may lead to an excessive proteostatic load that can no longer be handled by the existing folding capacity. Bottom: Minocycline targets the ribosome to attenuate translation, thus aligning proteostatic load with folding capacity, mimicking the effect of the ISR. The thickness of the 'translation' and 'folding' arrows are drawn to signify the relative capacity of each system.
DOI: https://doi.org/10.7554/eLife.40314.017

Mortison et al. arrived at the same MOA studying the effects of doxycycline on inflammation and tumor growth (*Garrido-Mesa et al., 2013*; *Mortison et al., 2018*).

Directly lowering translation through mutations, pharmacological agents or RNAi is a geroprotective mechanism that extends lifespan across taxa (*Kapahi et al., 2004*; *Kaeberlein et al., 2005*; *Hansen et al., 2007*; *Harrison et al., 2009*; *Selman et al., 2009*; *Han et al., 2017*). Our study reveals that the many beneficial effects of minocycline in eukaryotes are elicited by directly attenuating translation and that this mechanism is capable of reducing protein aggregation even when initiated in old, SSP-deficient *C. elegans* (*Figure 1G,H*). While the effect on longevity in old organisms is similar between minocycline and rapamycin, our epistasis analysis shows that minocycline differs with regards to some genetic requirements. Inhibition of mTOR signaling or S6 phosphorylation does not extend lifespan of TFEB/*hlh-30* mutants and requires the *hsf-1*-mediated heat-shock response to extend lifespan and to prevent protein aggregation (*Robida-Stubbs et al., 2012*; *Lapierre et al., 2013*; *Seo et al., 2013*). In contrast, minocycline extends lifespan of *hlh-30(tm1978)* and of *hsf-1 (sy441)* mutants (*Figure 2*) and prevents protein aggregation in *C. elegans* and human cells despite inhibiting the activation of the heat-shock response by *hsf-1* (*Figure 2B–D*; *Figure 4C–F*) (*Prahlad et al., 2008*; *Steinkraus et al., 2008*). These differences raise the question on how translation attenuation by minocycline extends longevity independently from SSPs that are required for other translation-related mechanisms. A likely explanation stems from the way minocycline attenuates translation. Any mechanism that targets the 18S rRNA, the catalytic site of the ribosome, must non-selectively attenuate translation. In contrast, many other well-investigated interventions target translation initiation, which in principle allows for selective attenuation of translation of subgroups of mRNAs. For example, knock down of eIF4G/*ifg-1* has been shown to lead to a general attenuation of translation but also to a selective increase in translation of SSP-related mRNAs, which play a role in the protective effect (*Rogers et al., 2011*). Thus, whether or not interventions lower translation selectively or non-selectively and thus involve SSPs is likely to depend on whether they target translation initiation or aspects of peptide bond formation. Previous studies used cycloheximide to show that blocking translation by interfering with the translocation step protects from protein aggregation, as newly synthesized proteins are the main protein species susceptible to damage and to collateral misfolding under stress. An obvious testable prediction of our minocycline data is that the protective effect of cycloheximide should also be independent of any SSPs (*Medicherla and Goldberg, 2008*; *Zhou et al., 2014*; *Xu et al., 2016*). If minocycline acts non-selectively, how is its effect greater on polysomes than on monosomes? Any single ribosome slowed down by minocycline will negatively affect all subsequent ribosomes on the same mRNA, thus amplifying the effect in a manner dependent on ribosome number. Future studies will be necessary to precisely elucidate these details.

Similar to minocycline, Smith et al. have shown that bacterial deprivation initiated in day-8-old *C. elegans* also extends lifespan, even as much as when initiating it at day 4 (*Smith et al., 2008*). Although not directly measured in this study, bacterial deprivation is also likely to dramatically reduce cytoplasmic translation, as suggested by *Steinkraus et al. (2008)*. From a therapeutic perspective, mechanisms that prevent protein aggregation independently of the activation of the HSF1 network are especially interesting as the presence of pathological protein aggregates during disease already suggests an overtaxed protein folding machinery (*Balch et al., 2008*).

Attenuation of translation is a core cellular protection mechanism. In response to stress, SSPs activate the ISR, which, through phosphorylation of eIF2$\alpha$, selectively attenuates translation (*Novoa et al., 2003*; *Martin et al., 2014*; *Pakos-Zebrucka et al., 2016*). Attenuation of translation provides the cell with a recovery period to restore proteostasis before translation resumes after dephosphorylating eIF2$\alpha$. Recovery after stress is marked by intense translation of chaperone mRNAs like HSP70 or BiP that involves the formation of polysomes, taxing the preexisting proteostasis network (*Novoa et al., 2003*; *Das et al., 2015*). Commencing translation after a stressful event, before proteostasis is restored, results in the production of ROS and caspase-3-mediated apoptosis (*Han et al., 2013*). By attenuating translation, minocycline prolongs the period for proteostasis to recover similar to what has been observed for the PPP1R15A inhibitor Sephin1 (*Das et al., 2015*). By preferentially reducing polysome formation (*Figure 4B*), minocycline generally reduces peak load on the proteostasis network while still allowing for sufficient translation to maintain cellular function. Our discovery of the MOA of minocycline confirms the previous predictions made by Han et al.

(*Han et al., 2013*) that limiting protein synthesis should protect from protein aggregation (*Choe et al., 2016*).

Our proposed MOA also explains why the geroprotective effect of minocycline remains inducible into later age compared to many other mechanisms. Longevity mechanisms like reduced insulin/IGF signaling elicit geroprotective effects through the activation of SSPs. As SSPs are no longer inducible in post-stress-responsive adults, their geroprotective effects can no longer be activated in older animals, excluding the possibility that minocycline extends lifespan when initiated in post-stress responsive adults by directly inhibiting mitochondrial translation and the subsequent activation of the mitochondrial UPR (*Dillin et al., 2002a*; *Dillin et al., 2002b*; *Baker et al., 2012*; *Labbadia and Morimoto, 2015*; *Rangaraju et al., 2015*). Minocycline bypasses the activation of SSPs or the ISR (*Figure 2*; *Figure 4*) to attenuate translation, eliciting geroprotective effects by lowering the concentration of newly synthesized, aggregation-prone proteins, aligning proteostatic load with the lower proteostasis network capacity in older adults (*Figure 6*) (*Balch et al., 2008*; *Plate et al., 2016*).

While it is not known whether minocycline extends lifespan in mammals, its geroprotective effects reduce age-associated protein aggregation and inflammation as evidenced by numerous preclinical and clinical studies (*Supplementary file 1*; *Supplementary file 2*). The multitude of seemingly unrelated effects has been used to argue that tetracyclines act by a wide variety of distinct MOAs. Excitement about several clinically relevant findings was dampened by the lack of a clear MOA necessary to design eukaryote specific derivatives of these drugs (*Metz et al., 2004*; *Lampl et al., 2007*). Our results shed new light on these observations. Translation attenuation by reducing ribosomal load as an MOA provides a simple and compelling explanation for these seemingly unrelated beneficial effects.

For example, minocycline reduces inflammation in peripheral tissues as well as in the central nervous system (*Ferretti et al., 2012*). Initiation of the inflammatory cascade is similar to the initiation of other stress responses and leads to the activation of the transcription factors NF-κB and AP1 that induce expression of pro-inflammatory cytokines such as interleukin-1 (IL-1), interleukin-6 (IL-6), TNF-α or nitric oxide synthase (NOS). While it has been previously suggested that the anti-inflammatory effects of minocycline are achieved by the inhibition of MMP9, translational attenuation of pro-inflammatory factors provides a compelling alternative. Similarly, the beneficial effects of minocycline in treating fragile X syndrome could also be explained by translational attenuation, as fragile X is caused by the failure to properly express the translational repressor FMRP (*Li et al., 2001*).

However, it is also important to note that minocycline showed some detrimental effects in a trial of amyotrophic lateral sclerosis (ALS) (*Gordon et al., 2007*). As expected from an antibiotic, minocycline-treated patients showed more adverse gastrointestinal effects such as nausea, diarrhea and constipation. Whether or not minocycline was detrimental for ALS patients was later questioned on the basis that the used dosage was too high, exacerbating the negative side effects (*Leigh et al., 2008*). Given that ALS patients show a hypermetabolic phenotype, the gastrointestinal adverse side effects may have masked beneficial effects (*Jésus et al., 2018*), making a minocycline-like compound that lacks the antibiotic activity highly desirable.

Repurposing FDA-approved drugs such as minocycline using phenotypic screens reveals promising effects outside the primary indication (antibiotic) of minocycline and inevitably leads to promising new drug target(s) and MOAs. Our data suggest that the antibiotic activity of minocycline compel a minocycline structure-activity relationship (SAR) campaign to improve eukaryotic translational attenuation while eliminating antibiotic and other activities. In summary, our studies on minocycline shed light on the plasticity of longevity mechanisms upon aging and reveal an MOA for minocycline that explains its geroprotective effects.

## Materials and methods

**Key resources table**

| Reagent type (species) or resource | Designation | Source or reference | Identifiers | Additional information |
|---|---|---|---|---|

*Continued on next page*

*Continued*

| Reagent type (species) or resource | Designation | Source or reference | Identifiers | Additional information |
|---|---|---|---|---|
| Strain, strain background (*Caenorhabditis elegans*) | N2 | Caenorhabditis Genetics Center | RRID:WB-STRAIN:N2_(ancestral) | wild-type (Bristol) |
| Strain, strain background (*C. elegans*) | CL2166 | CGC | RRID:WB-STRAIN:CL2166 | *dvIs19 [(pAF15)gst -4p::GFP::NLS] III* |
| Strain, strain background (*C. elegans*) | SJ4005 | CGC | RRID:WB-STRAIN:SJ4005 | *zcIs4 [hsp-4::GFP] V* |
| Strain, strain background (*C. elegans*) | SJ4100 | CGC | RRID:WB-STRAIN:SJ4100 | *zcIs13 [hsp-6::GFP]* |
| Strain, strain background (*C. elegans*) | CL2070 | CGC | RRID:WB-STRAIN:CL2070 | *dvIs70 [hsp-16.2p:: GFP + rol-6 (su1006)]* |
| Strain, strain background (*C. elegans*) | SJ17 | CGC | RRID:WB-STRAIN:SJ17 | *xbp-1(zc12) III; zcIs4 V* |
| Strain, strain background (*C. elegans*) | QV225 | CGC | RRID:WB-STRAIN:QV225 | *skn-1(zj15) IV* |
| Strain, strain background (*C. elegans*) | PS3551 | CGC | RRID:WB-STRAIN:PS3551 | *hsf-1(sy441) I* |
| Strain, strain background (*C. elegans*) | CF1038 | CGC | RRID:WB-STRAIN:CF1038 | *daf-16(mu86) I* |
| Strain, strain background (*C. elegans*) | CB369 | CGC | RRID:WB-STRAIN:CB369 | *unc-51(e369) V* |
| Strain, strain background (*C. elegans*) | RB1206 | CGC | RRID:WB-STRAIN:RB1206 | *rsks-1(ok1255) III* |
| Strain, strain background (*C. elegans*) | CB3388 | CGC | RRID:WB-STRAIN:CB3388 | *ncl-1(e1865) III* |
| Strain, strain background (*C. elegans*) | CL2006 | CGC | RRID:WB-STRAIN:CL2006 | *dvIs2(pCL12 (unc-54:hu-Aβ $_{1-42}$) +pRF4)* |
| Strain, strain background (*C. elegans*) | NL5901 | CGC | RRID:WB-STRAIN:NL5901 | *pkIs2386 [α-synuclein:: YFP unc-119(+)]* |
| Strain, strain background (*C. elegans*) | MAH93 | Other | | *glp-1(ar202), unc-119(ed3), ltIs38[pAA1; pie-1/GFP:: PH(PLCdelta1); unc-119 (+)] III; ltIs37[pAA64; pie-1/mCHERRY ::his-58, unc-119 (+)]IV*; gift from M. Hansen |
| Strain, strain background (*C. elegans*) | MAH686 | Other | | *hlh-30(tm1978) IV*; gift from M. Hansen |

*Continued on next page*

*Continued*

| Reagent type (species) or resource | Designation | Source or reference | Identifiers | Additional information |
|---|---|---|---|---|
| Strain, strain background (*C. elegans*) | CMH5 | DOI: 10.1371/journal.pone.0159989 | | *atfs-1(tm4525) V* |
| Strain, strain background (*C. elegans*) | BC20306 | Baillie Genome GFP Project, Simon Fraser University | RRID:WB:-STRAIN:BC20306 | *cyp-34A9::GFP* |
| Cell line (*Homo sapiens*) | HeLa | ATCC | Cat # CCL-2, RRID:CVCL_0030 | |
| Cell line (*Mus musculus*) | NIH 3T3 | ATCC | Cat # CRL-1658, RRID:CVCL_0594 | |
| Cell line (*H. sapiens*) | HEK293^DAX | DOI: 10.1016/j.celrep.2013.03.024 | | |
| Antibody | anti-actin (mouse monoclonal) | MP Biomedicals | Cat # 08691001, RRID:AB_2335127 | (1:500) |
| Antibody | anti-Hsp60 (mouse monoclonal) | ThermoFisher | Cat # MA3-012, RRID:AB_2121466 | (1:250) |
| Antibody | anti-Hsp70/72 (mouse monoclonal) | Enzo | Cat # ADI-SPA-810, RRID:AB_10616513 | (1:1000) |
| Antibody | anti-GRP78 (BiP) (rabbit polyclonal) | Abcam | Cat # ab21685, RRID:AB_2119834 | (1:1000, from 1 mg/ml) |
| Antibody | IRDye 800CW (secondary) | Li-Cor | Cat # 926–32210, RRID:AB_621842 | (1:10000) |
| Antibody | IRDye 800CW (secondary) | Li-Cor | Cat # 926–32211, RRID:AB_621843 | (1:10000) |
| Sequence-based reagent | *HSP60* forward primer, 5'-GCAGAGTTCCTCAGAAGTTGG-3' | DOI: 10.1186/s12974-016-0486-x | | qRT-PCR |
| Sequence-based reagent | *HSP60* reverse primer, 5'-GCATCCAGTAAGGCAGTTCTC-3' | DOI: 10.1186/s12974-016-0486-x | | qRT-PCR |
| Sequence-based reagent | *HSPA1A (HSP70)* forward primer, 5'-GGAGGCGGAGAAGTACA-3' | DOI: 10.1021/cb500062n | | qRT-PCR |
| Sequence-based reagent | *HSPA1A (HSP70)* reverse primer, 5'-GCTGATGATGGGGTTAACA-3' | DOI: 10.1021/cb500062n | | qRT-PCR |
| Sequence-based reagent | *hsp-16.1/.11* forward primer, 5'-ACCACTATTTCCGTCCAGCT-3' | DOI: 10.7554/eLife.08833 | | qRT-PCR |
| Sequence-based reagent | *hsp-16.1/.11* reverse primer, 5'-TGACGTTCCATCTGAGCCAT-3' | DOI: 10.7554/eLife.08833 | | qRT-PCR |
| Sequence-based reagent | *hsp-16.2* forward primer, 5'-TCGATTGAAGCGCCAAAGAA-3' | DOI: 10.7554/eLife.08833 | | qRT-PCR |
| Sequence-based reagent | *hsp-16.2* reverse primer, 5'-TCTCTTCGACGATTGCCTGT-3' | DOI: 10.7554/eLife.08833 | | qRT-PCR |

*Continued on next page*

*Continued*

| Reagent type (species) or resource | Designation | Source or reference | Identifiers | Additional information |
|---|---|---|---|---|
| Sequence-based reagent | *hsp-16.41* forward primer, 5'-TCTTGGACGAACTCACTGGA-3' | DOI: 10.7554/eLife.08833 | | qRT-PCR |
| Sequence-based reagent | *hsp-16.41* reverse primer, 5'-AGAGACATCGAGTTGAACCGA-3' | DOI: 10.7554/eLife.08833 | | qRT-PCR |
| Sequence-based reagent | *hsp-16.48/.49* forward primer, 5'-CTCATGCTCCGTTCTCCATT-3' | DOI: 10.7554/eLife.08833 | | qRT-PCR |
| Sequence-based reagent | *hsp-16.48/.49* reverse primer, 5'-GAGTTGTGATCAGCATTTCTCCA-3' | DOI: 10.7554/eLife.08833 | | qRT-PCR |
| Sequence-based reagent | GFP forward primer, 5'-GGTCCTTCTTGAGTTTGTAAC-3' | DOI: 10.1074/jbc.C100556200 | | qRT-PCR |
| Sequence-based reagent | GFP reverse primer, 5'-CTCCACTGACAGAAAATTTG-3' | DOI: 10.1074/jbc.C100556200 | | qRT-PCR |
| Sequence-based reagent | *SDHA* forward primer, 5'-TGGTGCTGGTTGTCTCATTA-3' | DOI: 10.1134/S0003683813090032 | | qRT-PCR |
| Sequence-based reagent | *SDHA* reverse primer, 5'-ACCTTTCGCCTTGACTGTT-3' | DOI: 10.1134/S0003683813090032 | | qRT-PCR |
| Sequence-based reagent | *HSPC3* forward primer, 5'-ATGGAAGAGAGCAAGGCAAA-3' | DOI: 10.1134/S0003683813090032 | | qRT-PCR |
| Sequence-based reagent | *HSPC3* reverse primer, 5'-AATGCAGCAAGGTGAAGACA-3' | DOI: 10.1134/S0003683813090032 | | qRT-PCR |
| Sequence-based reagent | *crn-3* forward primer, 5'-GAATGCACTCATGAACAAAGTC-3' | DOI: 10.7554/eLife.08833 | | qRT-PCR |
| Sequence-based reagent | *crn-3* reverse primer, 5'-TAATGTTCGACTGATGAACCG-3' | DOI: 10.7554/eLife.08833 | | qRT-PCR |
| Sequence-based reagent | *xpg-1* forward primer, 5'-ATTGAGAACAGGATCATGAGG-3' | DOI: 10.7554/eLife.08833 | | qRT-PCR |
| Sequence-based reagent | *xpg-1* reverse primer, 5'-ACTAGCAACTCGTTTATCATCC-3' | DOI: 10.7554/eLife.08833 | | qRT-PCR |
| Sequence-based reagent | *rpl-6* forward primer, 5'-ACTAGCAACTCGTTTATCATCC-3' | DOI: 10.7554/eLife.08833 | | qRT-PCR |

*Continued on next page*

*Continued*

| Reagent type (species) or resource | Designation | Source or reference | Identifiers | Additional information |
|---|---|---|---|---|
| Sequence-based reagent | *rpl-6* reverse primer, 5'-GACAGTCTTGGAATGTCCGA-3' | DOI: 10.7554/eLife.08833 | | qRT-PCR |
| Commercial assay or kit | 20S Proteasome Activity Assay Kit | Chemicon International | Cat # APT280 | |
| Commercial assay or kit | RNeasy Mini Kit | Qiagen | Cat # 74104 | |
| Commercial assay or kit | iScript RT-Supermix | Bio-Rad | Cat # 170–8841 | |
| Commercial assay or kit | SsoAdvanced SYBR Green Supermix | Bio-Rad | Cat # 172–5264 | |
| Commercial assay or kit | PROTEOSTAT Prot. aggregation assay | Enzo | Cat # ENZ-51023 | |
| Chemical compound, drug | minocycline | Mp Biomedicals | Cat # 0215571891 | |
| Chemical compound, drug | methyl viologen hydrate (paraquat) | Acros Organics | Cat # 227320010 | |
| Chemical compound, drug | tunicamycin | LKT Laboratories | Cat # T8153 | |
| Chemical compound, drug | thapsigargin | Cayman Chemical | Cat # 10522 | |
| Chemical compound, drug | ISRIB | R and D Systems | Cat # 5284/10 | |
| Chemical compound, drug | levamisole | Mp Biomedicals | Cat # 0215522810 | |
| Chemical compound, drug | sodium arsenite | Spectrum Chemical | Cat # S1135 | |
| Chemical compound, drug | cycloheximide | Alfa Aesar | Cat # J66901-03 | |
| Software, algorithm | CellProfiler | Broad Institute | RRID:SCR_007358 | α-synuclein::YFP aggregate number and worm size analyses |
| Software, algorithm | Bio-Rad CFX Manager | Bio-Rad | | qRT-PCR analysis |
| Software, algorithm | ImageQuant | GE Healthcare Life Sciences | RRID:SCR_014246 | 35S incorporation analysis |

## *C. elegans* strains

The Bristol strain (N2) was used as the wild-type strain. The following worm strains used in this study were obtained from the Caenorhabditis Genetics Center (CGC; Minneapolis, MN) unless otherwise noted. CL2166 [*dvIs19 [(pAF15)gst-4p::GFP::NLS]*], SJ4005 [*zcIs4 [hsp-4p::GFP]*], SJ4100 [*zcIs13 [hsp-6p::GFP]*], CL2070 [*dvIs70 [hsp-16.2p::GFP + rol-6(su1006)]*], SJ17 [*xbp-1(zc12); zcIs4*], QV225 [*skn-1 (zj15)*], PS3551 [*hsf-1(sy441)*], CF1038 [*daf-16(mu86)*], CB369 [*unc-51(e369)*], RB1206 [*rsks-1(ok1255)*], CB3388 [*ncl-1(e1865)*], CL2006 [*dvIs2(pCL12(unc-54:hu-Aβ $_{1–42}$)+pRF4)*], NL5901 [*pkIs2386 [α-synuclein::YFP unc-119(+)]*]. Strains were backcrossed at least three times prior to experimental analysis. MAH93 [*glp-1(ar202), unc-119(ed3), ltIs38[pAA1; pie 1/GFP::PH(PLCdelta1); unc-119 (+)]; ltIs37 [pAA64; pie-1/mCHERRY::his-58, unc-119 (+)]*] and MAH686 [*hlh-30(tm1978)*] were gifts from Malene Hansen and Caroline Kumsta and *atfs-1(tm4525) V* was a gift from Cole Haynes (*Nargund et al., 2012*). BC20306 [*cyp-34A9p::GFP*] was received from Baillie Genome GFP Project (Simon Fraser University, Burnaby, Vancouver, Canada).

## Worm stress imaging

1000 – 2000 age-synchronized L1 GFP reporter animals were plated into 6 cm culture plates with liquid medium (S-complete medium with 50 µg/ml carbenicillin and 0.1 µg/ml fungizone [Amphotericin B]) containing 6 mg/ml *Escherichia coli* OP50 ($1.5 \times 10^8$ colony-forming units [cfu]/ml, carbenicillin-resistant to exclude growth of other bacteria), freshly prepared 4 days in advance, as previously described (*Solis and Petrascheck, 2011*), and were maintained at 20°C. The final volume in each plate was 7 ml. To prevent self-fertilization, 5-fluoro-2′-deoxyuridine (FUDR, 0.12 mM final) (Sigma-Aldrich, cat # 856657) was added 42 – 45 hr after seeding. With day 1 being the first day of young adulthood (first day past the L4 stage), various stressors were added to each strain at different time points: 0.5 mM arsenite to day 1/day 5/day 8 CL2166 *gst-4p::GFP* and imaged 5 hr later, 0.5 mM paraquat to the late L4/day 4/day 7 stage of SJ4100 *hsp-6p::GFP* and imaged 24 hr later, 5 µg/ml tunicamycin to day 1/day 5/day 8 SJ4005 *hsp-4p::GFP* and imaged 8 hr later and a 1.5 hr heat shock at 35°C followed by recovery at 20°C to day 1/day 5/day 8 CL2070 *hsp-16.2p::GFP* and imaged 8 hr later. Worm GFP intensity was quantified using a COPAS FP BIOSORT (Union Biometrica) and animals were sorted into 96-well plates and imaged using an ImageXpress Micro XL High-Content screening system (Molecular Devices) with a 2x objective. To determine minocycline's affect on GFP expression, 100 µM minocycline was added 2 hr prior to each stressor at the day 4/day 5 stage.

## Lifespan assay

Age-synchronized *C. elegans* were prepared in liquid medium (S-complete medium with 50 µg/ml carbenicillin and 0.1 µg/ml fungizone) in flat-bottom, optically clear 96-well plates (Corning, cat # 351172) containing 150 µl total volume per well, as previously described (*Solis and Petrascheck, 2011*). Plates contained ~10 animals per well in 6 mg/ml OP50. All experiments with minocycline were prepared with γ-irradiated OP50. Age-synchronized animals were seeded as L1 larvae and grown at 20°C. Plates were covered with sealers to prevent evaporation. To prevent self-fertilization, FUDR (0.12 mM final) was added 42 – 45 hr after seeding. Drugs were added on the days indicated and survival was scored manually by visualizing worm movement using an inverted microscope 3x/week. When used, DMSO was kept to a final concentration of 0.33% v/v. Statistical analysis was performed using the Mantel–Haenszel version of the log-rank test as outlined in Petrascheck and Miller (*Petrascheck and Miller, 2017*).

## α-synuclein::YFP imaging

NL5901 (*pkIs2386 [α-synuclein::YFP unc-119(+)]*) age-synchronized animals were treated with water on day 1 or 100 µM minocycline on days 1, 5 or 8. On days 8, 11, 16 or 19, 10 – 15 water- and minocycline-treated animals of each treatment stage were transferred to glass slides containing 3% agarose pads and paralyzed by adding a drop of a 1 mM levamisole solution dissolved in M9 buffer. Brightfield and fluorescence images were taken with a 20x objective using the ImageXpress Micro XL. The number of α-synuclein aggregates in the pharynx region of each worm (25% of total body length) were determined by analyzing images using a custom pipeline created in CellProfiler software.

## CL2006 abeta paralysis

Day 1 CL2006 (*dvIs2(pCL12(unc-54:hu-Aβ$_{1-42}$)+pRF4)*) synchronized animals were treated with water or 100 µM minocycline. On day 4, 50 – 100 animals each were transferred to three NGM plates and placed at 37°C for 2 hr. After heat shock, animals were scored to determine the number of paralyzed animals. Animals were marked as paralyzed if they did not show mid-body movement upon light touch to the pharynx with a worm pick.

## Thermotolerance

Age-synchronized N2 animals were prepared in 6 cm culture plates and treated with water or 100 µM minocycline on day 1. On evening day 4, 30 – 50 animals each were transferred to 6 cm NGM plates in triplicate for each condition and heat stress was induced at 35°C. The first survival measurement was taken 8 hr later by lightly touching animals with a worm pick and scoring for movement. Plates were kept at 35°C and measurements of survival were taken every 2 hr until nearly all water-treated N2s were dead.

## Paraquat stress resistance assay

Resistance to oxidative stress was determined by measuring survival of untreated and minocycline-treated animals after a 24 hr exposure to the ROS-generator paraquat (methyl viologen hydrate; Acros Organics, cat # 227320010). Experimental *C. elegans* cultures were set up as described in Lifespan Assay. Water or 100 µM minocycline was added on day 1. Paraquat was added to a final concentration of 0, 15, 25, 50, 75 and 100 mM on day 5 of adulthood and survival of animals was assessed 24 hr after paraquat addition. It's important to note, we only saw protection from paraquat-induced death when using dead, γ-irradiated OP50 and did not see protection when using live OP50.

## Iodoacetamide isoTOP-ABPP

20,000 age-synchronized N2 *C. elegans* were grown in liquid culture in 15 cm culture plates (Corning, cat # 351058), two plates per condition. Animals were treated with water or 100 µM minocycline on day 5. Animals were collected on day 8, washed 3x with cold DPBS (Gibco, cat # 14190 – 136) and flash-frozen in liquid nitrogen. 50 µl 1.4 mm zirconium oxide beads (Precellys, cat # 03961-1-103) and 50 µl 0.5 mm glass beads (Precellys, cat # 03961-1-104) were added to each sample and animals were lysed using a Precellys 24 tissue homogenizer. Bradford assay (Bio-rad, cat # 5000002) was used to determine protein concentration for each sample for normalization. Proteome sample (1 mg) was treated with 100 mM IA-alkyne probe by adding 5 ml of a 10 mM probe stock (in DMSO). The labeling reactions were incubated at room temperature for 1 hr, after which the samples were conjugated to isotopically labeled, TEV-cleavable tags (TEV tags) by copper-catalyzed azide-alkyne cycloaddition (CuACC or 'click chemistry'). Heavy click chemistry reaction mixture (60 ml) was added to the water-treated control sample, and light reaction mixture (60 ml) was added to the minocycline-treated sample. The click reaction mixture consisted of TEV tags [10 ml of a 5 mM stock, light (minocycline-treated) or heavy (water treated)], CuSO4 (10 ml of a 50 mM stock in water), and tris (benzyltriazolylmethyl) amine (30 ml of a 1.7 mM stock in 4:1 tBuOH/DMSO), to which tris(2- carboxyethyl)phosphine (10 ml of a 50 mM stock) was added. The reaction was performed for 1 hr at room temperature. The light- and heavy-labeled samples were then centrifuged at 16,000 x g for 5 min at 4°C to harvest the precipitated proteins. The resulting pellets were resuspended in 500 ml of cold methanol by sonication, and the heavy and light samples were combined pairwise. Combined pellets were then washed with cold methanol, after which the pellet was solubilized by sonication in DPBS with 1.2% SDS. The samples were heated at 90°C for 5 min and subjected to streptavidin enrichment of probe-labeled proteins, sequential on-bead trypsin and TEV digestion, and liquid chromatography–tandem mass spectrometry.

RAW Xtractor (version 1.9.9.2; available at http://fields.scripps.edu/ downloads.php) was used to extract the MS2 spectra data from the raw files (MS2 spectra data correspond to the fragments analyzed during the second stage of mass spectrometry). MS2 data were searched against a reverse concatenated, nonredundant variant of the *C. elegans* UniProt database (release, 2017_09) with the ProLuCID algorithm (publicly available at http://fields.scripps.edu/downloads.php). Cysteine residues were searched with a static modification for carboxyamidomethylation (+57.02146) and up to one differential modification for either the light or heavy TEV tags (+464.28595 or+470.29976, respectively). Peptides were required to have at least one tryptic terminus and to contain the TEV modification. ProLuCID data were filtered through DTASelect (version 2.0) to achieve a peptide false-positive rate below 1%.

The quantification of light/heavy ratios (isoTOP-ABPP ratios, R values) was performed by in-house CIMAGE software using default parameters (three MS1s per peak and a signal-to-noise threshold of 2.5). See *Figure 3—source data 1* for additional filtering parameters.

## Offspring assay

N2 animals were age-synchronized and allowed to grow in liquid medium in a 6 cm plate. By the L4 stage, a single worm was transferred into each well in the first column of a 96-well plate (eight animals) containing liquid medium without FUDR, two plates per condition. On day 1, animals were treated with water or 100 µM minocycline. Every 24 hr, the adult animals were transferred to the well in the next column over and progeny was scored from the previous well using an inverted microscope. This process was continued until day 5 of adulthood.

## Worm size assay

Animals prepared as in the lifespan assay and treated with water or 100 µM minocycline on day 1 were imaged across 4 days using a 10x objective with the ImageXpress Micro XL and the length in pixels of each worm measured was determined using CellProfiler analysis software.

## Metabolic mass spectrometry

Animals and bacteria were metabolically labeled with the desired nitrogen isotope ($^{14}$N or $^{15}$N) as previously described (*Gomez-Amaro et al., 2015*). After a minimum of three generations, ~20,000 age-synchronized larvae per condition were transferred to 15 cm culture dishes and cultured in liquid medium plus metabolically labeled $^{14}$N (experimental) or $^{15}$N (mixed-population standard) bacteria (6 mg/ml). Water or 100 µM minocycline were added and animals were collected at different time points (L4→day 1, day 5→day 6, day 5→day 8). To control for differences in protein content due to minocycline treatment, samples were normalized by RNA concentration.

Lysed animals were prepared for mass spectrometry by precipitation in 13% trichloroacetic acid (TCA) overnight at 4°C. The protein pellet was collected by centrifugation at maximum speed for 20 min. The pellet was washed with 10% TCA and spun again for 10 min at 4°. The pellet was then washed with ice cold acetone and spun for an additional 10 min at 4°. The protein pellet was resuspended in 100 – 200 µl of 100 mM ammonium bicarbonate with 5% (v/v) acetonitrile. Then 10% (by volume) of 50 mM DTT was added and the sample was incubated for 10 min at 65°. This incubation was followed by the addition of 10% (by volume) of 1 mM iodoacetic acid (IAA) and a 30 min incubation at 30° in the dark. A total of 5 µg of trypsin was added to each sample and incubated overnight at 37°. Trypsinized samples were then cleaned up for mass spectrometry using PepClean columns (Pierce) following the manufacturer's directions. Clean samples were dried in a speed vacuum and then resuspended in 10 µl of 5% acetonitrile with 0.1% (v/v) formic acid. Samples were spun at high speed to remove particulates before placing in mass spectrometry tubes for analysis.

Samples were analyzed on an ABSCIEX 5600 Triple time of flight (TOF) mass spectrometer coupled to an Eksigent nano-LC Ultra equipped with a nanoflex cHiPLC system. Conditions on the ABSCIEX 5600 were as follows: The source gas conditions were optimized for each experiment and were generally set to GS1 = 8–12, GS2 = 0, and curtain gas = 25. The source temperature was set to 150°. Information dependent acquisition experiments were run with a 2 s cycle time, with 0.5 ms accumulation time in the MS1 scan and up to 20 candidate ions per MS/MS time period. The m/z range in the MS1 scans was 400 – 1250 Da, and 100 – 1800 Da for the MS/MS scans. Target ions were excluded after two occurrences for 12 s to increase sequencing coverage. Target ions with +2 to+5 charge were selected for sequencing once they reached a threshold of 100 counts per second.

Conditions for the Eksigent nano-HPLC were as follows: Gradients were run with a trap-and-elute setup with water plus 0.1% (v/v) formic acid as the mobile phase A and acetonitrile with 0.1% (v/v) formic acid as mobile phase B. Samples were loaded onto a 200 µm × 0.5 mm ChromXP C18-CL 3 µm 120 Å Trap column at 2 µl/min. Gradients were run from 5% mobile phase A to 40% mobile phase B over 2 hr on a 75 µm × 15 cm ChromXP C-18-CL 3 µm 120 Å analytical column. This was followed by a rapid jump to 80% B for 10 min to clean the column and a 20 min reequilibration at 95% A. Water sample blanks were run between samples to rid the column of any residual interfering peptides, including a short gradient, followed by the 80% B and 20 min reequilibration with 5% A.

Data were converted using the ABSCIEX conversion software to mgf format and MZML format. The peak list was generated by searching a SwissProt database using MASCOT, with the taxonomy set to *C. elegans* and *E. coli* simultaneously (Matrix Science). An MS/MS Ion search was performed using the $^{15}$N quantification, with a peptide mass tolerance of ±0.1 Da and a fragment mass tolerance of ±0.1 Da. The maximum number of missed cleavages was set to 2. MS1 scans for identified peptides were fit to three isotope distributions using ISODIST.

## 20S proteasome assay

2500 age-synchronized animals were grown on 10-cm culture plates (Corning, cat # 351029) per condition in liquid medium. Water or 100 µM minocycline were added on day 1. On day 5, animals were collected and washed 3x with cold DPBS, then once with 1x assay buffer (Chemicon International, cat # APT280). Animals were lysed with Precellys homogenizer and spun down at 12,000 x g for 5 min. Protein concentrations were measured and normalized using Bradford assay. 20S proteasome

activity was measured for each lysate according to the protocol (Chemicon International, cat # APT280). 200 µg of samples were loaded with assay mixture into each well of a UV-transparent 96-well plate and incubated for 1 hr at 37°C. Fluorescence was measured on the Tecan Safire II with a 380/460 nm filter set.

## Cell culture authentication and testing

HeLa cells (ATCC CCL-2) or NIH 3T3 cells (ATCC CRL-1658) were directly obtained from ATCC. Mycoplasma testing was done every 6 months through TSRI. The identity of the HEK293$^{DAX}$ cells was verified by the induction of ATF6 through TMP.

## $^{35}$S Incorporation

HeLa cells (ATCC CCL-2) or NIH 3T3 cells (ATCC CRL-1658) were grown at 37°C and plated onto a six-well plate (Corning, cat # 353046) in DMEM (Life Technologies, cat # 11995073) supplemented with 10% FBS (ATCC, cat # 30 – 2021) and penicillin-streptomycin and the indicated concentrations of cycloheximide or minocycline were added to 70% confluent cells and incubated for 6 hr. DMEM culture media was removed and wells were washed 2x with warm DPBS. 2 ml of a DMEM solution without methionine or cysteine (Life Technologies, cat # 21013024) supplemented with 10% dialyzed FBS (VWR Intl., cat # 101301 – 494) and [$^{35}$S]-methionine/cysteine (PerkinElmer, cat # NEG772002MC) was added to each well for 1 hr. Media was removed and wells were washed 2x with cold DPBS and 200 µl of cold RIPA buffer (Fisher Scientific, cat # 507513771) with protease inhibitor (cOmplete mini EDTA-free, Sigma-Aldrich, cat # 11836170001) was added to each well for 15 min on ice. Lysates were collected and spun down at 5000 g for 10 min. Samples normalized by protein concentration, determined by Bradford assay, were run on an SDS-PAGE gel. The gel was stained with Coomassie Brilliant Blue G-250 (MP Biomedicals, cat # 0219034325), imaged for loading control, dried and exposed overnight to a phosphor screen and imaged again using a Typhoon 9400 imager. Lanes were quantified using ImageQuant software (GE Healthcare Life Sciences).

## Polysome profile

HeLa cells were grown on 10 cm culture plates (Olympus Plastics, #25 – 202) at 37°C in DMEM culture media supplemented with 10% FBS and penicillin-streptomycin and water or 100 µM minocycline was added to 70% confluent cells for 12 hr. 100 µg/ml cycloheximide was added to each plate for 10 min. On ice, medium was removed, plates were washed 2x with cold wash buffer (DPBS supplemented with 100 µg/ml cycloheximide) and cells were removed in 1 ml wash buffer using a cell scraper. Cells were transferred to eppendorf tubes and spun down for 5 min at 2300 g at 4°C. Supernatant was removed and 300 µl cold hypotonic buffer (1.5 mM KCl, 10 mM MgCl2, 5 mM Tris-HCl, pH 7.4, 100 µg/ml cycloheximide) was added and gently mixed. 300 µl cold hypotonic lysis buffer (2% sodium deoxycholate, 2% Triton X-100, 2.5 mM DTT, 100 µg/ml cycloheximide, and 10 units of RNAsin/ml in hypotonic buffer) was added and cells were homogenized by using 10 strokes with a glass dounce homogenizer on ice. Samples were spun down at 2300 x g for 10 min at 4°C. Supernatants were transferred to fresh eppendorf tubes and RNA concentration was determined by measuring the A$_{260}$. 2 mg RNA were loaded onto 17 – 51% sucrose gradients prepared with a gradient maker in SW41 tubes (Denville, cat # U5030). Tubes were centrifuged at 40,000 rpm for 2.5 hr at 4°C using an SW41 rotor. OD$_{260}$ was measured for each sample using an online Isco Model UA-5 Absorbance/Fluorescence Monitor.

## HSP western blot

HeLa cells were grown on 10 cm culture plates at 37°C in DMEM supplemented with 10% FBS and penicillin-streptomycin and water or 100 µM minocycline was added for 12 hr to 70% confluent cells. Cells were transferred to 43°C for a 1 hr heat shock, then returned to 37°C for a 6 hr recovery period. Plates were washed 2x with cold DPBS, then 500 µl cold RIPA buffer containing protease inhibitor was added for 15 min on ice. Lysates were collected, normalized by RNA concentration, run on an SDS-PAGE gel and transferred for immunoblot detection. Anti-actin (MP Biomedicals, cat # 08691001), anti-Hsp60 (ThermoFisher, cat # MA3-012) and anti-Hsp70/72 (Enzo, cat # ADI-SPA-810) primary antibodies and IRDye 800CW secondary antibody (Li-Cor, cat # 926 – 32210) were used for detection.

## Quantitative real-time PCR (qRT-PCR) and data analysis

All qRT-PCR experiments were conducted according to the MIQE guidelines (*Bustin et al., 2009*), except that samples were not tested in a bio-analyzer, but photometrically quantified using a Nanodrop. HeLa cells were grown as described above to 70% confluency and treated with or without 100 µM minocycline, two 10 cm plates per condition. 12 hr after treatment, one each of the untreated and minocycline-treated plates were subjected to a 1 hr heat shock at 43°C and returned to 37°C for a 3 hr recovery. Cells were collected and RNA was extracted as described in the Qiagen RNeasy mini kit protocol (Qiagen, cat # 74104). Cell culture plates were washed two times with warm DPBS, followed by addition of Buffer RLT. Cells were removed from plates with a cell scraper and collected in eppendorf tubes. RNeasy spin columns were used to purify RNA and on-column DNase digestion was performed. For *C. elegans* samples,~2000 age-synchronized day 1 CL2070 *hsp-16.2p::GFP* animals cultured in 10 cm plates were treated with water or 100 µM minocycline for 2 hr prior to inducing heat shock at 35°C for 1.5 hr, and harvested immediately after. Harvested animals were washed three times in ice cold DPBS and frozen in liquid nitrogen. To extract RNA, frozen animals were resuspended in ice-cold Trizol (Qiagen, cat # 79306), zirconium beads, and glass beads in the ratio of 5:1:1 respectively, and disrupted in Precellys lysing system (6500 rpm, 3 × 10 s cycles) followed by chloroform extraction. RNA was precipitated using isopropanol and washed once with 75% ethanol followed by DNAse (Sigma-Aldrich, cat # AMPD1-1KT) treatment.

For all RNA samples, reverse transcription was carried out using iScript RT-Supermix (Bio-Rad, cat # 170 – 8841) at 42°C for 30 min. Quantitative PCR reactions were set up in 384-well plates (Bio-Rad, cat # HSP3901), which included 2.5 µl Bio-Rad SsoAdvanced SYBR Green Supermix (cat # 172 – 5264), 1 µl cDNA template (2.5 ng/µl, to final of 0.5 ng/µl in 5 µl PCR reaction), 1 µl water, and 0.5 µl of forward and reverse primers (150 nM final concentration). Quantitative PCR was carried out using a Bio-Rad CFX384 Real-Time thermocycler (95°C, 3 min; 40 cycles of 95°C 10 s, 60°C 30 s; Melting curve: 95°C 5 s, 60–95°C at 0.5°C increment, 10 s). Gene expression was normalized to two references genes for the HeLa cell samples, SDHA and HSPC3, and three reference genes for *C. elegans* samples, *crn-3*, *xpg-1* and *rpl-6*, using the Bio-Rad CFX Manager software. Statistical significance was determined using Student's t-test

## HEK293$^{DAX}$western blot

HEK293$^{DAX}$ cells were grown on 6-well plates at 37°C in DMEM supplemented with 10% FBS and penicillin-streptomycin. Water or 100 µM minocycline was added with or without 20 uM trimethoprim for 12 hr to 70% confluent cells. Plates were washed 2x with cold DPBS, then 200 µl cold RIPA buffer containing protease inhibitor was added for 15 min on ice. Lysates were collected, normalized by RNA concentration, run on an SDS-PAGE gel and transferred for immunoblot detection. Anti-actin and anti-GRP78 BiP primary (Abcam, cat # ab21685) antibodies and IRDye 800CW (Li-Cor, anti-mouse, cat # 926 – 32210 and anti-rabbit, cat # 926 – 32211) secondary antibodies were used for detection.

## ProteoStat aggregation assay

~70% confluent HeLa cells plated on two 6-well plates were treated for 6 hr with water or 100 µM minocycline, three wells per treatment. One plate was heat shocked for 1 hr at 43°C followed by a 1 hr recovery period at 37°C. Wells were washed two times with cold DPBS and then incubated on ice for 15 min with cold RIPA +protease inhibitor. Lysate for each well was collected and spun down at 5000 x g for 10 min. Protein concentration was measured for each sample using and all samples were normalized. Aggregation was measured using the PROTEOSTAT detection reagent (Enzo, cat # ENZ-51023) for 10 µg protein of each sample. Fluorescence was measured with excitation of 550 nm and emission at of 600 nm on a Tecan Sapphire two plate reader over a 30-min period, with one reading per minute. Fluorescence values for each sample were averaged over the 30-min period to determine a corresponding aggregate value normalized to the water treated samples.

## ISRIB translation

HeLa cells were plated onto a 6-well plate in DMEM culture media supplemented with 10% FBS and penicillin-streptomycin and ISRIB (200 nM) (R and D Systems, cat # 5284/10). Minocycline (100 µM), thapsigargin (200 nM) (Cayman Chemical, cat # 10522) or different combinations of the three were

added to 70% confluent cells and incubated for 3 hr (ISRIB and/or thapsigargin-treated samples) or 6 hr (minocycline-treated samples). $^{35}$S incorporation was measured using the protocol described above ($^{35}$S incorporation).

## Yeast lifespan assay

Yeast RLS assays were performed by isolating virgin daughter cells of the BY4741 strain and then allowed to grow into mother cells while their corresponding daughters were microdissected and counted, until the mother cell could no longer divide. Data were analyzed using the Wilcoxon rank-sum test.

## Acknowledgements

We are grateful to Lars Plate and Kelly Rainbolt for technical help, the Hansen lab for the MAH93 and MAH686 strains, Luke Wiseman for experimental advice, HEK293$^{DAX}$ cells and $^{35}$S isotope, and Jeffery Kelly, Malene Hansen, Anabel Perez, and Alan To for critical comments to the manuscript. This work was supported by grants to M.P from the NIH (DP2 OD008398, R21NS107951), the Ellison Foundation (AG-NS-0928 – 12) and the Glenn Foundation and a fellowship to G.S from the NSF (NSF/DGE-1346837).

## Additional information

### Competing interests

Benjamin F Cravatt: Reviewing editor, *eLife*. The other authors declare that no competing interests exist.

### Funding

| Funder | Grant reference number | Author |
|---|---|---|
| Lawrence Ellison Foundation | AG-NS-0928-12 | Michael Petrascheck |
| National Institutes of Health | DP2 OD008398 | Michael Petrascheck |
| National Science Foundation | NSF/DGE-1346837 | Gregory M Solis |
| National Institutes of Health | R21NS107951 | Michael Petrascheck |
| Glenn Foundation for Medical Research | | Michael Petrascheck |

The funders had no role in study design, data collection and interpretation, or the decision to submit the work for publication.

### Author contributions

Gregory M Solis, Conceptualization, Data curation, Software, Formal analysis, Supervision, Funding acquisition, Validation, Investigation, Visualization, Methodology, Writing—original draft, Project administration, Writing—review and editing; Rozina Kardakaris, Formal analysis, Investigation, Project administration, Writing—review and editing; Elizabeth R Valentine, Liron Bar-Peled, Data curation, Formal analysis, Validation, Investigation; Alice L Chen, Megan M Blewett, Mark A McCormick, Formal analysis, Validation, Investigation; James R Williamson, Brian Kennedy, Benjamin F Cravatt, Resources, Supervision; Michael Petrascheck, Conceptualization, Resources, Data curation, Formal analysis, Supervision, Funding acquisition, Validation, Investigation, Visualization, Methodology, Writing—original draft, Project administration, Writing—review and editing

### Author ORCIDs

Gregory M Solis [iD] http://orcid.org/0000-0002-3347-5980
James R Williamson [iD] http://orcid.org/0000-0002-8772-468X
Michael Petrascheck [iD] http://orcid.org/0000-0002-1010-145X

**Decision letter and Author response**
Decision letter https://doi.org/10.7554/eLife.40314.022
Author response https://doi.org/10.7554/eLife.40314.023

## Additional files

### Supplementary files
• Supplementary file 1. Select minocycline clinical trials.
DOI: https://doi.org/10.7554/eLife.40314.018

• Supplementary file 2. Completed and ongoing clinical trials with minocycline.
DOI: https://doi.org/10.7554/eLife.40314.019

• Transparent reporting form
DOI: https://doi.org/10.7554/eLife.40314.020

### Data availability
All data regarding the presented evidence are part of this manuscript. Datasets that were used to make any figures are associated as supplementary files.

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
