## [Decision Letter]

[**Editorial note:** This article has been through an editorial process in which the authors decide how to respond to the issues raised during peer review. The Reviewing Editor's assessment is that all the issues have been addressed.]

Thank you for submitting your article "Translation attenuation by minocycline enhances longevity and proteostasis in old post-stress-responsive organisms" for consideration by *eLife*. Your article has been reviewed by three peer reviewers, including Matt Kaeberlein as the Reviewing Editor and Reviewer #1, and the evaluation has been overseen by Philip Cole as the Senior Editor. The following individuals involved in review of your submission have also agreed to reveal their identity: Aric Rodgers (Reviewer #2) and Jan Gruber (Reviewer #3).

The Reviewing Editor has highlighted the concerns that require revision and/or responses, and we have included the separate reviews below for your consideration. If you have any questions, please do not hesitate to contact us.

Summary:

All of the reviewers are quite enthusiastic about the importance of this work and quality of the data. No major concerns were identified, although several minor concerns about interpretation and overstating the data were noted. The authors are encouraged to consider these referee comments and revise the text accordingly.

Separate reviews (please respond to each point):

*Reviewer #1:*

Solis et al. present a compelling model for the late life pro-longevity effects of minocycline: that it acts to selectively reduce translation of highly expressed, aggregation prone proteins, thereby delaying proteostatic collapse in aged cells and animals. The experiments appear to be well-conducted and there are several interesting and cool observations here that move the field forward. I have only relatively minor suggestions.

Minor Comments:

1) It is mentioned that CR and rapamycin can both extend lifespan when initiated late in life in mice. It would be worth noting in the Discussion that this has also been shown to be true for bacterial deprivation in *C. elegans* (Smith et al., *Caenorhabditis elegans*2008) where initiating food deprivation at day 8 extends lifespan as much as initiating the treatment at day 4 or day 2. This could conceivably work via a similar mechanism to minocycline as complete food deprivation is almost certain to dramatically reduce cytoplasmic translation.

2) It is perhaps an overstatement to say that minocycline "did not result in an increased susceptibility to stress", when this is only shown for heat stress. Were other stressors tested? If not, then perhaps qualify this statement a bit.

3) Figure 2—figure supplement 1C is missing the legend.

4) The interpretation of the autophagy experiments may also be a bit overstated. Unless you are working with autophagy null mutations, the fact that minocycline can still extend these mutants can't be definitively interpreted. Perhaps better to say that the data are consistent with the model that the MOA for minocycline does not act through autophagy. If you really want to strengthen this, you could look for induction of autophagy via a variety of methods.

5) I would soften the statement in the Discussion that the "MOA of minocycline is clearly distinct from other interventions targeting translation". While I agree that the genetic epistasis data support this model, they do not rule out the possibility that there are overlapping but partially distinct mechanisms at play. It is also possible that some interventions require HSF-1 and/or HLH-30 for indirect reasons without those factors being causally downstream in the mechanism of lifespan extension.

6) I would be careful about saying that mechanisms like "the UPRmt elicit geroprotective effects through the activation of SSPs". Multiple labs have now shown that the UPRmt (or at least induction of all the factors commonly referred to as the UPRmt) is not in fact geroprotective. Indeed, the UPRmt (or at least induction of all the factors commonly referred to as the UPRmt) is neither necessary nor sufficient to extend lifespan. I realize that the Dillin and Auwerx labs choose to ignore these data and propagate their own "alternative facts" rather than address the gaping holes in their model, but perhaps you want to pick a different example stress response pathway for this particular phrase.

*Reviewer #2:*

Results in Figures 1E and 2B are quite remarkable and the study is of high interest. Greatest concerns are about interpretation, with a couple of minor issues regarding study design:

From the Abstract, the authors ".. propose that minocycline lowers the concentration of newly synthesized aggregation-prone proteins, resulting in a relative increase in protein-folding capacity without the necessity to induce protein-folding pathways." It is easy to rationalize that translation suppression, itself, is the protective mechanism important for mitigating damage from unfolded proteins by allowing chaperones to focus on maintaining conformation of, or eliminating, pre-existing proteins. This sounds very nice, but there is little evidence because translation is inhibited during stress regardless of addition of translational suppressors. So, what is the mechanism? Yes, minocycline reduces translation, but how certain are the authors that its translational suppression, per se, that is protective? Is it possible that mino helps prevent aggregation directly? Perhaps this could be tested in vitro? Its ability to extend lifespan in a sick (*hsf-1*) mutant is especially provocative. Is its function somehow bypassing the need for some HSF-1 gene targets? The possibility that the mechanism involved is simply preventing de novo synthesis of aggregation-prone proteins is not convincing. Where is the evidence for that? Select changes in protein synthesis (i.e., differential translation) pointed to selective depletion of factors for ribosome biosynthesis (Figure 3) as well as other highly translated factors according to polysome status (Figure 4). Is that what is meant by "aggregation prone proteins?" I suppose that is one way of characterizing them (and a *lot* of other factors). All of this should be more thoroughly argued in the manuscript.

Results show near-instantaneous protection (2 hour treatment with minocycline), so why measure proteome after 3 days of treatment? If protection is conferred so rapidly, what is the point at reading the proteome 3 days after treatment? Rapid (2 hour) protection from unfolded protein stress is highly unlikely to be proteome based unless it is through post-translational modification (e.g., phosphorylation). This should be more thoroughly addressed in the manuscript.

Minor Comments:

Given that minocycline is inhibiting translation, it may be better to measure transcription of stress response pathway genes directly as opposed to using transcriptional reporters (Figure 1), which depend on translation to produce fluorescence.

"growth" is misspelled, end of the first paragraph of the Discussion.

It is important to consider differential translation and de novo synthesis when doing western blots to look at changes in HSPs in a system that suppresses total cellular translation. Alternative methods of measuring de novo synthesis should be considered.

"Minocycline is a tetracycline antibiotic regulatory agency-approved for the treatment of acne". Odd sentence, consider revising.

*Reviewer #3:*

This, in my opinion, is really exciting and important work. The approach of specifically targeting post-stress response animals for lifespan extension is elegant and, importantly, worked to identify what looks like a drug with a very unique MOA.

The amount and quality of work done to explore this MOA and to identify targets is impressive and the result appears quite conclusive (although I am not an expert on some of the techniques involved).

I do have some very minor comments, suggestions and questions (see below) – however, none of these substantially impact the main conclusions.

My comments mainly pertain to minor issues of interpretation or emphasis. In summary, I think this is an impressive piece of work and I am quite excited about it.

Minor Comments:

It is true that the upshift from 20 °C to 25 °C triggers the Ab phenotype of the CL2006 strain, but I think this is a specific interaction between the Ab1-42 peptide and temperature and I do not think that it is completely correct to equate this with an effect on general heat shock induced protein aggregation.

Similarly, the authors only show that minocycline has a protective effect on one of the SSP tested (heat shock) but the discussion makes it sound like this is a general effect (That SSP responses are blunted but survival under challenge is improved nonetheless).

Unless I am mistaken, only the first part (blunting) is demonstrated for all four challenges while improved survival is only shown for heat shock. That being said, I understand why heat shock was chosen (the attenuation of the GFP response is most convincing and so the fact that there is a protective effect is most impressive).

The authors emphasize the lifespan effect in the *hsf-1* background. While this is an interesting observation with an impressive lifespan extension that, at least in terms of% , is indeed larger than in WT, the *hsf-1* strain, even when treated, still appears shorter lived than WT (or, at best, identical to untreated WT). The% effect is large, precisely because this mutant is so short lived.

I think based on these data alone it is hard to determine if minocycline mainly acts to rescue *hsf-1* or rescues the mutation and extends lifespan (as suggested). On a related note, the lifespan benefits in *skn-1* and maybe even *daf-16* background, while statistically maybe not as large (at least in terms of percent extension), seem even more surprising and impressive to me (in that these animals are more long-lived that WT controls).

In the Discussion, it may be worthwhile to re-iterate the observation that rapamycin also has benefits in aged animals and relate this to the role of *rsks^-1^* on translation downstream of mTOR.

Also, looking at minocycline and its use in humans, in addition to the protective / beneficial effects listed in Table 1, there are some reports of detrimental effects, especially in the context of some auto-immune diseases. Maybe it would be worth discussing evidence (if any) whether these may be related to the effect on translation, its antibiotic efficacy or are completely unrelated. I appreciate that nothing may be known about this, but it strikes me as a point worth touching on when discussing translational potential.

It may be worth comparing the doses used here (e.g. in tissue culture) to typical therapeutic concentrations in humans (e.g. in plasma).

---

## [Author Response]

Reviewer #1:Solis et al. present a compelling model for the late life pro-longevity effects of minocycline: that it acts to selectively reduce translation of highly expressed, aggregation prone proteins, thereby delaying proteostatic collapse in aged cells and animals. The experiments appear to be well-conducted and there are several interesting and cool observations here that move the field forward. I have only relatively minor suggestions.Minor Comments:1) It is mentioned that CR and rapamycin can both extend lifespan when initiated late in life in mice. It would be worth noting in the Discussion that this has also been shown to be true for bacterial deprivation in C. elegans (Smith et al., 2008) where initiating food deprivation at day 8 extends lifespan as much as initiating the treatment at day 4 or day 2. This could conceivably work via a similar mechanism to minocycline as complete food deprivation is almost certain to dramatically reduce cytoplasmic translation.

Thank you for the suggestion. We have added the following sentences to the Discussion:

“Similar to minocycline, Smith et al. have shown that bacterial deprivation initiated in day-8-old *C. elegans* also extends lifespan, even as much as when initiating it at day 4 (Smith et al., 2008). Although not directly measured in this study, bacterial deprivation is also likely to dramatically reduce cytoplasmic translation, as suggested by Steinkraus et al. (Steinkraus et al., 2008).”

2) It is perhaps an overstatement to say that minocycline "did not result in an increased susceptibility to stress", when this is only shown for heat stress. Were other stressors tested? If not, then perhaps qualify this statement a bit.

To address this point, which was also raised by reviewer #3 (second point), we have added additional data. We have now included an experiment showing minocycline protects from paraquat-induced oxidative stress despite blocking the stress reporter response to paraquat. After 24 hour 100mM PQ treatment, 75% of 100 μm minocycline-treated animals are still alive compared to 44% in control animals. We have also provided additional references in the literature with examples of minocycline protecting from paraquat-induced cell death.

Amended text inserted into the Results section:

“Furthermore, minocycline-treated adults were also much more resistant to paraquat-induced oxidative stress (Figure 2—figure supplement 1D). This is consistent with the observation that minocycline also protects neuronal-like rat pheochromocytoma (PC12) cells from paraquat-induced cell death (Huang et al., 2012).”

3) Figure 2—figure supplement 1C is missing the legend.

We have now added a legend in-between both panel C and D (new figure).

4) The interpretation of the autophagy experiments may also be a bit overstated. Unless you are working with autophagy null mutations, the fact that minocycline can still extend these mutants can't be definitively interpreted. Perhaps better to say that the data are consistent with the model that the MOA for minocycline does not act through autophagy.

To remedy our overstatement we changed the text describing the results as follows:

“Despite the severe behavioral and morphological phenotypes observed in the *unc-51(e369)* mutants, the mutation did not reduce the ability of minocycline to extend lifespan. Minocycline also extended the lifespan of *hlh-30(tm1978)* mutants, but less than the 45% observed in N2. These genetic results, and the previous finding that activation of autophagy in already old *C. elegans* is likely to be (Wilhelm et al., 2017) are consistent with a mechanisms of minocycline that acts in part independently of the activation of autophagy. However, as *unc-51* is essential, the residual activity remaining in the *unc-51(e369)* mutants does not allow us to draw a definitive conclusion.”

5) I would soften the statement in the Discussion that the "MOA of minocycline is clearly distinct from other interventions targeting translation". While I agree that the genetic epistasis data support this model, they do not rule out the possibility that there are overlapping but partially distinct mechanisms at play. It is also possible that some interventions require HSF-1 and/or HLH-30 for indirect reasons without those factors being causally downstream in the mechanism of lifespan extension.

All three reviewers had similar concerns regarding the relation of the mechanism of minocycline to other translation-related mechanisms. After some considerations we agree that minocycline is likely to act on a mechanism that overlaps with previous mechanisms. The observed differences are likely the result of whether translation is inhibited during the initiation step or during the elongation/translocation steps during peptide bond formation. Targeting the initiation step can conceivably lead to selective inhibition of translation that allows translation of stress response factors as shown for *igf-1* while targeting the peptide bond formation with drugs such as minocycline or cycloheximide will be non-selective. We have replaced the previous section with a longer, and hopefully more nuanced explanation. We amended the passage as follows:

“While the effect on longevity in old organisms is similar between minocycline and rapamycin, our epistasis analysis shows that minocycline differs with regards to some genetic requirements. Inhibition of mTOR signaling or S6 phosphorylation does not extend lifespan of TFEB/*hlh-30* mutants and requires the *hsf-1*-mediated heat shock response to extend lifespan and to prevent protein aggregation (Robida-Stubbs et al., 2012; Lapierre et al., 2013; Seo et al., 2013). In contrast, minocycline extends lifespan of *hlh-30(tm1978)* and of *hsf-1(sy441*) mutants (Figure 2)and prevents protein aggregation in *C. elegans* and human cells despite inhibiting the activation of the heat shock response by *hsf-1* (Figure 2B-D; Figure 4C-F) (Prahlad et al., 2008; Steinkraus et al., 2008). These differences raise the question on how translation attenuation by minocycline extends longevity independently from SSPs that are required for other translation-related mechanisms. A likely explanation stems from the way minocycline attenuates translation. Any mechanism that targets the 18S rRNA, the catalytic site of the ribosome, must non-selectively attenuate translation. In contrast, many other well-investigated interventions target translation initiation which in principle allows for selective attenuation of translation of subgroups of mRNAs. For example, knock down of eIF4G/ *ifg-1* has been shown to lead to a general attenuation of translation but also to a selective increase in translation of SSP-related mRNAs, which play a role in the protective effect (Rogers et al., 2011). Thus, whether or not interventions lower translation selectively or non-selectively and thus involve SSPs is likely to depend on whether they target translation initiation or aspects of peptide bond formation. Previous studies used cycloheximide to show that blocking translation by interfering with the translocation step protects from protein aggregation, as newly synthesized proteins are the main protein species susceptible to damage and to collateral misfolding under stress. An obvious testable prediction of our minocycline data is that the protective effect of cycloheximide should also be independent of any SSPs (Medicherla and Goldberg, 2008). If minocycline acts non-selectively, how is its effect greater on polysomes than on monosomes? Any single ribosome slowed down by minocycline will negatively affect all subsequent ribosomes on the same mRNA, thus amplifying the effect in a manner dependent on ribosome number. Future studies will be necessary to precisely elucidate these details.”

6) I would be careful about saying that mechanisms like "the UPRmt elicit geroprotective effects through the activation of SSPs". Multiple labs have now shown that the UPRmt (or at least induction of all the factors commonly referred to as the UPRmt) is not in fact geroprotective. Indeed, the UPRmt (or at least induction of all the factors commonly referred to as the UPRmt) is neither necessary nor sufficient to extend lifespan. I realize that the Dillin and Auwerx labs choose to ignore these data and propagate their own "alternative facts" rather than address the gaping holes in their model, but perhaps you want to pick a different example stress response pathway for this particular phrase.

We have addressed this concern by amending the sentence as follows:

“Longevity mechanisms like reduced insulin/IGF signaling elicit geroprotective effects through the activation of SSPs.”

Reviewer #2:

Results in Figures 1E and 2B are quite remarkable and the study is of high interest. Greatest concerns are about interpretation, with a couple of minor issues regarding study design:From the Abstract, the authors ".. propose that minocycline lowers the concentration of newly synthesized aggregation-prone proteins, resulting in a relative increase in protein-folding capacity without the necessity to induce protein-folding pathways." It is easy to rationalize that translation suppression, itself, is the protective mechanism important for mitigating damage from unfolded proteins by allowing chaperones to focus on maintaining conformation of, or eliminating, pre-existing proteins. This sounds very nice, but there is little evidence because translation is inhibited during stress regardless of addition of translational suppressors. So, what is the mechanism? Yes, minocycline reduces translation, but how certain are the authors that its translational suppression, per se, that is protective?

Because reviewer #1 had a related concern we have tried to incorporate much of the answers following below into a new and hopefully more nuanced paragraph in the revised manuscript. See reviewer #1 point 5. The major point that we have to concede that we cannot knock out, or even mutate the 18S rRNA (there are over 100 copies), to show that when missing, minocycline fails to protect. We have increased the 18 rRNA level using the *nCl^-^1* mutant to show that the dose response curve shifts to the left. This constitutes a quantitative prediction which few models are able to make. Our current, long-term strategy is to generate new compounds that attenuate translation like minocycline but are non-antibiotic, but this is well beyond the scope of the current work.

Translational suppression per se is protective: We completely agree that stress reduces translation and that this is a major part of the cells protective response. We respectfully disagree on the notion that there is little evidence that inhibition of translation per se improves protein folding. The following papers show that newly synthesized proteins are the ones most susceptible to damage and protein aggregation, and thus, that reducing de-novo synthesis per se will reduce protein aggregation.

The earliest paper we are aware of is from A. Goldberg’s group that shows that stressors like heat or paraquat lead to the preferential degradation of newly-synthesized proteins that are less than 1 hour old. If degradation of newly-synthesized proteins upon stress is prevented, they aggregate. Older proteins (> 1 hr old), despite being measurably damaged by fore example paraquat, do not tend to aggregate. This finding reveals that for a protein, being newly-synthesized contributes more to the propensity to aggregate than being damaged. Pretreatment with the translation inhibitor cycloheximide prevents the synthesis of new proteins and thus protein aggregation, clearly showing that inhibition of translation per se prevents protein aggregation. (Medicherla and Goldberg, 2008). Later, Borchelt’s group came to the same conclusions using in mammalian cells (Xu et al., 2016) by showing again that pretreating HEK293 and other cells with cycloheximide prevents protein aggregation. These results cannot be explained by any model requiring a selective translation of, for example, chaperone mRNAs, as cycloheximide indiscriminately blocks translation directly within the ribosome and because only newly synthesized proteins are affected. These studies provide clear evidence that newly synthesized proteins are the ones mostly prone to aggregation and that reducing protein synthesis per se is protective. To extend the data regarding minocycline we now provided a new figure that shows that minocycline not only increases resistance to heat but also to paraquat as was shown in the Goldberg study. Protection from paraquat is achieved despite the data in Figure 2 that show that minocycline inhibits the oxidative stress response and the response to paraquat directly. It is also remarkable that minocycline reduces protein aggregation, despite also reducing proteasome activity, which clearly should increase the formation of aggregates.

While this has not been shown in the above cited papers, our data would predict that cycloheximide would also block activation of the heat shock and oxidative stress responses. Work by Rong Li’s lab goes one step further and suggests that reducing protein synthesis not only protects from protein aggregation by reducing the folding load but that protein aggregation requires new polypeptide synthesis (Zhou et al., 2014). Additional studies to that effect come from Bertolotti and colleagues (Das et al., 2015) and Kaufman and colleagues (Han et al., 2013). These previous studies by multiple different labs clearly confirm the notion that inhibition of translation per se is protective.

Selective vs. non-selective inhibition of translation: As we outlined in the new paragraph (see reviewer #1 point 5), a likely explanation on how the longevity genetics of interventions that target translation differ, is that targeting initiation factors can lead to a selective attenuation of translation as shown for *ifg-1*, while targeting the peptide forming machinery directly does not (minocycline, cycloheximide). Interventions directly targeting the peptide-forming activity within the ribosome, as we suggest for minocycline, cannot depend on the activation of a stress response, as the stress response factors will not be expressed at the protein level. Any model that requires the induction of a pathway and hence protein synthesis can be excluded as the mechanism of action for inhibitors that directly target the catalytic site of the ribosome.

We agree however that it is counterintuitive that inhibition of translation per se directly at the ribosome is protective. While this can be explained easily for short-term interventions like the use of cycloheximide, long-term use is certainly more problematic. We show that a well tolerated FDA-approved drug that is known to have beneficial effects in aggregation diseases in humans in clinical trials acts by such a mechanism and that directly targeting the ribosome may be a valuable therapeutic strategy, provided that it attenuates translation and does not shut it down completely.

Is it possible that mino helps prevent aggregation directly? Perhaps this could be tested in vitro?

In principle this could be tested, but for what protein? To prevent heat shock-induced protein aggregation for all proteins, as shown in Figure 4F, one would have to postulate that minocycline selectively binds any misfolded protein and then restores folding. The much simpler explanation, especially since reviewer #2 agrees that minocycline attenuates translation, is that minocycline prevents protein aggregation in the same manner as was shown for cycloheximide, by preventing de novo protein synthesis as newly synthesized proteins are the ones most prone to aggregation.

Its ability to extend lifespan in a sick (hsf-1) mutant is especially provocative. Is its function somehow bypassing the need for some HSF-1 gene targets? The possibility that the mechanism involved is simply preventing de novo synthesis of aggregation-prone proteins is not convincing. Where is the evidence for that? Select changes in protein synthesis (i.e., differential translation) pointed to selective depletion of factors for ribosome biosynthesis (Figure 3) as well as other highly translated factors according to polysome status (Figure 4). Is that what is meant by "aggregation prone proteins?" I suppose that is one way of characterizing them (and a lot of other factors). All of this should be more thoroughly argued in the manuscript.Results show near-instantaneous protection (2 hour treatment with minocycline), so why measure proteome after 3 days of treatment? If protection is conferred so rapidly, what is the point at reading the proteome 3 days after treatment? Rapid (2 hour) protection from unfolded protein stress is highly unlikely to be proteome based unless it is through post-translational modification (e.g., phosphorylation). This should be more thoroughly addressed in the manuscript.

We have re-written a part of the Discussion (see also reviewer #1 point 5), highlighting previous findings that show that blocking translation protects from protein aggregation. In mammalian cells, even a 30 min pretreatment with cycloheximide confers the same near instantaneous protection from protein aggregation as we see with minocycline because newly-synthesized proteins are the protein species most susceptible to damage, misfolding and protein aggregation (Medicherla and Goldberg, 2008; Xu et al., 2016). Blocking protein synthesis per se by cycloheximide is protective because it is the newly synthesized proteins that have a much larger propensity to aggregate. Minocycline and cycloheximide target the catalytic center of the ribosome, making it difficult to envision how such an effect could lead to selective translation. The pronounced effect on polysomes is caused by a collateral effect in which a single slow ribosome slows all subsequent ribosomes thus amplifying any effect in a manner dependent on the number for ribosomes per mRNA.

Different timings: To test if minocycline generally inhibits translation we used a 2 hr incubation (Figure 2, GFP and Figure 4, ^35^S) to mirror image the experiments previously done with cycloheximide. However, in Figure 3 we also considered the existence of selective translation and whether there are proteins whose translation is not inhibited by minocycline. Even if changes in translation are instantaneous, to be able to detect changes using proteomics depends on translation and degradation rates and thus the differences will take much longer to become detectable. Because of the near proteome-wide reduction of translation, which could have been the result of a normalization problem, a serious difficulty regarding all experiments in which translation is inhibited globally, we asked if the effect of minocycline depends on the translation rates of the animals. From our previous work and work by others we knew that translation rates are substantially reduced by day 5 of adulthood. Thus, if minocycline acts on translation, it must take longer to detect effects on the proteome in older animals (low translation) than in younger animals (high translation), which is what we observed. It is also noteworthy that the few proteins that show higher levels in minocycline-treated animals, like RPS-15, are negatively associated with lifespan, in that they extend lifespan when their expression is lowered (Rodrigues et al., 2012;). The three proteasome-related factors that showed an increase in expression had no functional consequence since measuring proteasomal activity was reduced (Figure 3—figure supplement 1). While this needs experimental confirmation, the higher levels of these factors are more likely to be result of the reduced proteasome activity resulting from minocycline treatment. To clarify this, we have added the following paragraph to the manuscript:

“These data were largely inconsistent with a model in which minocycline selectively reduced translation, as has been shown for *ifg-1*. The few factors that showed higher expression levels in minocycline-treated animals had no known link to stress resistance or were shown to increase lifespan when knocked down by RNAi.”

Minor Comments:Given that minocycline is inhibiting translation, it may be better to measure transcription of stress response pathway genes directly as opposed to using transcriptional reporters (Figure 1), which depend on translation to produce fluorescence.

Thank you for this point. In the previous version of the manuscript we have only done this for the cell culture experiments but in the revised version we have now measured transcriptional levels of a panel of heat shock proteins. These are all activated in response to heat shock as shown in the new Figure 4—figure supplement 1C. While we have not verified the expression of these factors upon heat shock on the protein level, the proteomic data suggest that in the un-induced state all “hsp-” proteins are expressed at a lower level in minocycline-treated animals than in untreated animals.

"growth" is misspelled, end of the first paragraph of the Discussion.

Thanks for the catch, corrected.

It is important to consider differential translation and de novo synthesis when doing western blots to look at changes in HSPs in a system that suppresses total cellular translation. Alternative methods of measuring de novo synthesis should be considered.

Yes, we agree. We attempted to identify differentially-translated components in the proteomics experiments in Figure 3, which revealed a general reduction of translation but provided no evidence for selective translation. The coverage of these experiments was limited and a riboseq profile will certainly provide a much wider coverage. However, our results showing that minocycline acts independently of any of the known major stress response transcription factors strongly argues against the selective translation of stress response factors and clearly favors the model in which reduced translation per se is protective.

"Minocycline is a tetracycline antibiotic regulatory agency-approved for the treatment of acne". Odd sentence, consider revising.

We have revised the sentence as follows:

“Minocycline is a regulatory agency-approved tetracycline antibiotic used to treat acne vulgaris and has long been known to reduce tumor growth, inflammation and protein aggregation in mammals by an unknown MOA.”

Reviewer #3:

This, in my opinion, is really exciting and important work. The approach of specifically targeting post-stress response animals for lifespan extension is elegant and, importantly, worked to identify what looks like a drug with a very unique MOA.The amount and quality of work done to explore this MOA and to identify targets is impressive and the result appears quite conclusive (although I am not an expert on some of the techniques involved).I do have some very minor comments, suggestions and questions (see below) – however, none of these substantially impact the main conclusions.My comments mainly pertain to minor issues of interpretation or emphasis. In summary, I think this is an impressive piece of work and I am quite excited about it.Minor Comments:It is true that the upshift from 20 °C to 25 °C triggers the Ab phenotype of the CL2006 strain, but I think this is a specific interaction between the Ab1-42 peptide and temperature and I do not think that it is completely correct to equate this with an effect on general heat shock induced protein aggregation.

We have amended the paragraph as follows:

“To confirm this effect extended beyond α-synuclein aggregation, we tested another strain model of protein aggregation and determined minocycline also reduced the paralysis caused by temperature-induced Aβ_1-42_ misfolding that leads to aggregation in *C. elegans*’body-wall muscle (Figure 1—figure supplement 1G) (Jiang et al., 2012; McColl et al., 2012). Thus, minocycline treatment reduced both age-dependent and temperature-induced protein aggregation in *C. elegans*.”

Similarly, the authors only show that minocycline has a protective effect on one of the SSP tested (heat shock) but the discussion makes it sound like this is a general effect (That SSP responses are blunted but survival under challenge is improved nonetheless).Unless I am mistaken, only the first part (blunting) is demonstrated for all four challenges while improved survival is only shown for heat shock. That being said, I understand why heat shock was chosen (the attenuation of the GFP response is most convincing and so the fact that there is a protective effect is most impressive).

We have now also tested the resistance to paraquat (Figure 2—figure supplement 1D). See Reviewer #1 (point 2).

The authors emphasize the lifespan effect in the hsf-1 background. While this is an interesting observation with an impressive lifespan extension that, at least in terms of% , is indeed larger than in WT, the hsf-1 strain, even when treated, still appears shorter lived than WT (or, at best, identical to untreated WT). The% effect is large, precisely because this mutant is so short lived.I think based on these data alone it is hard to determine if minocycline mainly acts to rescue hsf-1 or rescues the mutation and extends lifespan (as suggested). On a related note, the lifespan benefits in skn-1 and maybe even daf-16 background, while statistically maybe not as large (at least in terms of percent extension), seem even more surprising and impressive to me (in that these animals are more long-lived that WT controls).

We apologize because there has been a mistake in the data representation on our part. When assembling the paper, our table (Figure 2—source data 1) showed a 59%, 81% and 100% increase in lifespan for *hsf-1*. We therefore decided to show the middle, representative experiment in Figure 1 (+81%). However, the table should have said +159%, + 81% and +100%. The previous Figure 1 in the paper therefore shows the lowest lifespan extension observed in *hsf-1* mutants. In the amended version we show the correct middle representative experiment of a +100% increase. The absolute lifespans of minocycline-treated *hsf-1* mutants are 23 days, 31days and 32 days, clearly longer than the absolute lifespan of untreated N2 and for the latter two experiments close to even minocycline-treated N2 animals. This is a rather dramatic extension of lifespan in the presence of a mutation that shortens lifespan across many other longevity paradigms. Given the observation that minocycline reduces protein aggregation, the most likely cause for the short lifespan of *hsf-1* mutants, it seemed sensible to suggest a partial rescue. We agree however that this is somewhat speculative and hence we used the word “suggest” rather than “show” in our text. The passage was amended as follows:

“Remarkably, minocycline extended lifespan in *hsf-1(sy441)* mutants by up to +159%, a relative extension clearly greater than what was observed in wild-type N2 animals and almost reaching absolute lifespans of minocycline-treated N2 animals (Figure 1E; Figure 2—source data 1). This enhanced extension suggests that minocycline not only extends lifespan as in N2 but also rescues some of the defects associated with mutated *hsf-1(sy441)*.”

In the Discussion, it may be worthwhile to re-iterate the observation that rapamycin also has benefits in aged animals and relate this to the role of rsks^-1^ on translation downstream of mTOR.

Please see our revised paragraph in the Discussion starting with:

“While the effect on longevity in old organisms is similar between minocycline and rapamycin, our epistasis analysis…”

Also, looking at minocycline and its use in humans, in addition to the protective / beneficial effects listed in Table 1, there are some reports of detrimental effects, especially in the context of some auto-immune diseases. Maybe it would be worth discussing evidence (if any) whether these may be related to the effect on translation, its antibiotic efficacy or are completely unrelated. I appreciate that nothing may be known about this, but it strikes me as a point worth touching on when discussing translational potential.

Thank for this point. As an example for problematic effects we chose to discuss the ALS trials with minocycline that actually showed some detrimental effects:

However, it is also important to note that minocycline showed some detrimental effects in a trial of amyotrophic lateral sclerosis (ALS) (Gordon et al., 2007). As expected from an antibiotic, minocycline-treated patients showed more adverse gastrointestinal effects such as nausea, diarrhea and constipation. Whether or not minocycline was detrimental for ALS patients was later questioned on the basis that the used dosage was too high, exacerbating the negative side effects (Leigh et al., 2008). Given that ALS patients show a hypermetabolic phenotype, the gastrointestinal adverse side effects may have masked beneficial effects (Jesus et al., 2018), making a minocycline-like compound that lacks the antibiotic activity highly desirable.”

It may be worth comparing the doses used here (e.g. in tissue culture) to typical therapeutic concentrations in humans (e.g. in plasma).

The following sentence was added to the manuscript:

“Recording dose response curves we determined an EC_50_ of 22 μM and an optimal concentration of 50-100 μM, as compared to therapeutic concentrations measured in patient serum of 5-10 μM (Figure 1F; Figure 1—figure supplement 1E) (Agwuh and MacGowan, 2006; Garrido-Mesa et al., 2013).”